# On Fairness of Unified Multimodal Large Language Model for Image Generation

Ming Liu[1]*    Hao Chen[2]    Jindong Wang[3]
Liwen Wang[1]    Bhiksha Raj Ramakrishnan[2]    Wensheng Zhang[1]
[1]Iowa State University    [2]Carnegie Mellon University    [3]William & Mary

## Abstract

Unified multimodal large language models (U-MLLMs) have demonstrated impressive performance in end-to-end visual understanding and generation tasks. However, compared to generation-only systems (e.g., Stable Diffusion), the unified architecture of U-MLLMs introduces new risks of propagating demographic stereotypes. In this paper, we benchmark several state-of-the-art U-MLLMs and show that they exhibit significant gender and race biases in the generated outputs. To diagnose the source of these biases, we propose a *locate-then-fix* framework: we first audit the vision and language components — using techniques such as linear probing and controlled generation — and find that the language model appears to be a primary origin of the observed generative bias. Moreover, we observe a "partial alignment" phenomenon, where the U-MLLMs exhibit less bias in understanding tasks yet produce substantially biased images. To address this, we introduce a novel *balanced preference loss* that enforces uniform generation probabilities across demographics by leveraging a synthetically balanced dataset. Extensive experiments show that our approach significantly reduces demographic bias while preserving semantic fidelity and image quality. Our findings underscore the need for targeted debiasing strategies in unified multimodal systems and introduce a practical approach to mitigate biases.

## 1   Introduction

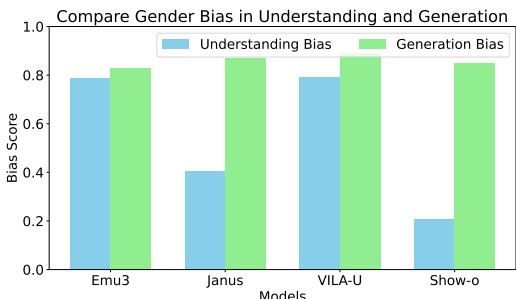
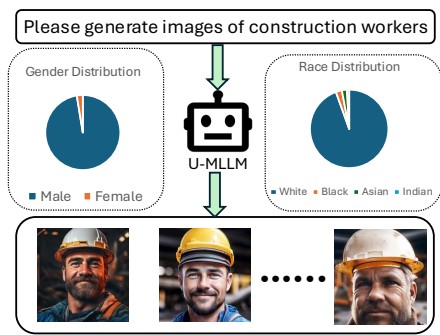

Figure 1: Biases in U-MLLMs. Left: Comparison between understanding and generation biases in the model. Right: U-MLLMs generate high-quality images but lack diversity, showing bias toward certain demographics (e.g., the model predominantly generates images of white males).

Multimodal large language models (MLLMs) have demonstrated remarkable capabilities in visual understanding [24, 39]. Recent research [42, 43] has focused on extending MLLMs' capabilities

---

*pkulium@iastate.edu

39th Conference on Neural Information Processing Systems (NeurIPS 2025).

to image generation settings, enabling them to produce both textual and *visual* content. These *unified* MLLMs (U-MLLMs), e.g., VILA-U [43], present both visual understanding and generation capability. They can not only understand the semantics in images but also generate high-quality images conditioning on user prompts in natural language. However, these U-MLLMs with unified capabilities might inadvertently reproduce or amplify *biases* at deployment, including gender and racial stereotypes embedded in their large-scale training data [7, 10].

A common structure in U-MLLMs is an image *tokenizer* that transforms images into discrete tokens through an *encoder-decoder* framework [11]. Specifically, the vision encoder compresses the input image into latent embeddings and then quantizes them into discrete tokens. Afterward, a decoder reconstructs the image from these tokens. This discrete tokenization bridges textual and visual modalities by analogizing image tokens to words in a sentence, enabling a single autoregressive objective that unifies text and image generation. While this design has proven effective in terms of quality and scalability, it also opens additional avenues for bias to propagate from the tokenizer.

Existing work on debiasing in image generation has highlighted the social risks posed by skewed output distributions, and various methods have been proposed to reduce bias in image generation [9, 4, 41, 13, 37]. Nevertheless, many such methods are designed specifically for diffusion models, which leverage different principles for image generation [34, 13]. As U-MLLMs with autoregressive image generation capabilities become increasingly prevalent, it is imperative to evaluate their biases in image generation and develop new methods to reduce biases in these token-based generation models. Moreover, it remains an open question whether the generation biases emerge more from the vision encoder, which generates image tokens, or from the language modeling component, which generates image tokens according to the given text prompt.

This paper focuses on investigating and mitigating demographic biases in U-MLLMs for *text-to-image* generation. Specifically, we take the first step to study gender and race biases for U-MLLMs. We benchmark the state-of-the-art U-MLLMs for gender and race biases using datasets and metrics introduced in a recent study on image generation fairness [34]. These models include VILA-U [43], Show-o [44], Janus [42], Janus-Pro [8], TokenFlow [32], Emu3 [40]. Our results show that these models exhibit notable gender and race biases in image generation (see an example in Figure 1). Next, we conduct a detailed audit of the vision encoder/decoder and the language model component to localize the source(s) of these biases, where we find that the biases are mainly from the language model. Finally, we synthesize high-quality training data with a balanced demographic distribution and propose a novel balanced preference loss to mitigate generation biases, inspired by recent research on direct preference optimization [33, 15].

Through extensive experiments with various U-MLLMs, we demonstrate that our approach significantly reduces biases (e.g., over-generation of certain genders or races) without sacrificing the quality of image generation. For example, for the VILA-U model, our method reduces its gender bias by 71.9% and increases the inception score by 12.2%. In summary, our key contributions are as follows.

- **Benchmarking Bias**: We benchmark the state-of-the-art U-MLLMs on race and gender bias and find that they exhibit biases to varying degrees. Notably, Janus-Pro, one of these U-MLLMs, exhibited the worst gender biases with a value of 0.90, compared to the Stable Diffusion with a bias value of 0.67.

- **Localizing Bias**: We inspect different components in the VILA-U model (vision encoder and language model) by using methods such as linear probing in image embedding space to pinpoint potential sources of biases and find that the biases are likely from the language model component.

- **Mitigating Bias**: We used the diffusion model to synthesize training data with a balanced demographic distribution. We also introduce a balanced preference loss inspired by direct preference optimization, with the objective of balancing the likelihood of visual generation towards the different demographic groups. We empirically show that our approach yields substantial improvement in demographic fairness while preserving the quality of image generation, thus providing a practical framework for developing unified MLLMs with greater fairness.

## 2 Preliminary

**Structure of U-MLLMs.** We consider an autoregressive U-MLLM that, given a textual prompt $x$, first converts $x$ into a sequence of text tokens $x_1, \ldots, x_{T_x}$ and then generates a sequence of image

tokens $z_1, \ldots, z_{T_z}$, which an image decoder can reconstruct into a final image $y$ (see Figure 5 for the pipeline). Let $\theta = (\theta_v, \theta_l)$ denote the model parameters, where $\theta_v$ is the image tokenizer (comprising an encoder and decoder) that converts input images into discrete tokens and decodes tokens back into images, and $\theta_l$ is the language model (LM) that processes and generates token sequences (both text and image tokens) in a unified autoregressive manner. As shown in Figure 6, the image encoder $E_{\theta_v}$ maps an image $y$ to latent embeddings, then quantizes them into a discrete token sequence $z_1, \ldots, z_{T_z}$. Conversely, the image decoder $D_{\theta_v}$ inverts this process, reconstructing an image from a given sequence of image tokens:

$$\{z_1, \ldots, z_{T_z}\} = E_{\boldsymbol{\theta}_v}(y), \tag{1}$$

$$y = D_{\boldsymbol{\theta}_v}(z_1, \ldots, z_{T_z}). \tag{2}$$

Meanwhile, $LM_{\theta_l}$ treats text and image tokens uniformly under a single next-token probability distribution:

$$P_{\boldsymbol{\theta}}(z_t \mid x, z_{<t}) = LM_{\boldsymbol{\theta}_l}(z_{t-1}, \ldots, z_1; x). \tag{3}$$

This design allows the U-MLLM to perform visual understanding by mapping an image into a semantic token space (via eq. (1)) and feeding those tokens into the LM, and to perform image generation by autoregressively sampling image tokens via eq. (3) and reconstructing an image from the sampled tokens via eq. (2).

**Demographic Bias.** When the model is prompted with neutral text (i.e., no explicit demographic specified), it may exhibit *demographic bias* by generating images skewed toward certain groups (e.g., mostly males or mostly a particular race). Figure 1 illustrates this: given the prompt "Please generate images of construction workers," a U-MLLM produces mostly images of white males. Formally, let $d \in D = \{d_1, \ldots, d_K\}$, where $D$ is a set of demographic labels (e.g., $D = \{\text{male}, \text{female}\}$ for gender or $\{\text{Asian}, \text{Black}, \text{White}, \text{Indian}\}$ for race). We must clarify that these labels are strictly used for benchmark compatibility [34]. We recognize that the gender categories represent **perceived gender presentation** in generated images, not an individual's gender identity, and this framework does not account for gender-diverse individuals. Similarly, the racial categories cannot represent the full spectrum of human diversity, including mixed-race individuals. Our goal is to diagnose and mitigate stereotypical associations within this benchmark framework.

For a neutral prompt $x$ (e.g., "a portrait of a construction worker"), an unbiased model would generate a set of images $\{y_1, y_2, \ldots\}$ whose demographic labels are balanced (e.g., 50/50 split by gender or uniform across races). By contrast, we find that the latest U-MLLMs often produce outputs where

$$P_\theta(\mathcal{C}(y) = d_i \mid x) \gg P_\theta(\mathcal{C}(y) = d_j \mid x) \tag{4}$$

for some demographics $d_i, d_j$ (e.g., $d_i$ = male, $d_j$ = female) when $x$ is neutral. Here $C(y)$ is a pretrained attribute classifier that labels each generated image $y$ with a demographic attribute $\hat{d}$. Such output indicates a strong demographic bias in demographic preferences. Our goal is to mitigate this bias while preserving overall image fidelity.

**Direct Preference Optimization** As an efficient alternative to RLHF [30], Direct Preference Optimization (DPO) [33] re-parameterizes rewards via the policy itself. Given a policy $\pi_\theta$ and reference $\pi_{\text{ref}}$, DPO defines the implicit reward:

$$r(x, y) = \beta \log \frac{\pi_\theta(y \mid x)}{\pi_{\text{ref}}(y \mid x)} + \beta \log Z(x), \tag{5}$$

where $Z(x)$ is a normalization constant, and models preference between outputs $y_w$ and $y_l$ with a Bradley–Terry formulation:

$$p(y_w \succ y_l \mid x) = \sigma\big(r(x, y_w) - r(x, y_l)\big). \tag{6}$$

DPO maximizes this preference probability directly, avoiding a separate reward model.

Reference-free methods [15, 29] such as ORPO [15] set

$$\text{odds}_\theta(y \mid x) = \frac{p_\theta(y \mid x)}{1 - p_\theta(y \mid x)}, \quad \text{OR}_\theta(y_w, y_l) = \frac{\text{odds}_\theta(y_w \mid x)}{\text{odds}_\theta(y_l \mid x)} \tag{7}$$

and employ the loss

$$\mathcal{L}_{\text{OR}} = -\log \sigma\big(\log \text{OR}_\theta(y_w, y_l)\big) \tag{8}$$

By encouraging a large odds ratio $\text{OR}_\theta(y_w, y_l)$, the model is pushed to *prefer* the response $y_w$ over $y_l$ *directly*, without relying on a separate "reference" model. We adapt these preference-optimization ideas to formulate our balanced preference loss for U-MLLM debiasing.

## 3 Locating Bias

As shown in Figure 2, to determine where demographic biases arise in U-MLLM, we examine its intermediate outputs. In particular, we analyze the sequence of image tokens produced by the language model (LM) and the latent image embeddings produced by the vision encoder. We consider two main hypotheses for the origin of bias: (1) bias could emerge from the LM's token generation process, or (2) bias could stem from the vision encoder–decoder pipeline.

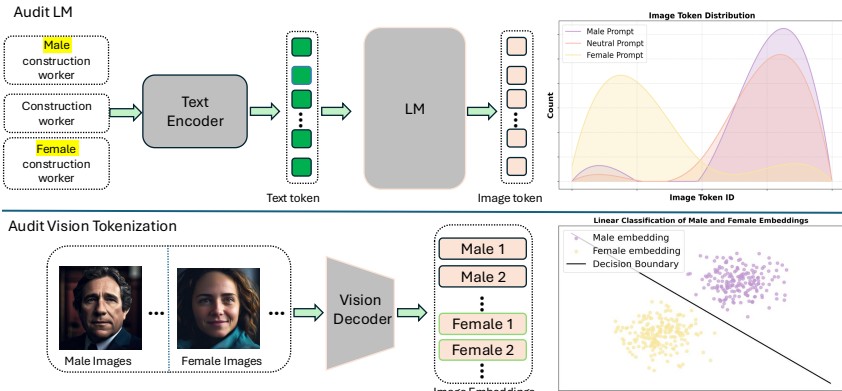

Figure 2: Detecting bias in LM (top), Vision tokenizer (bottom).

### 3.1 Hypothesis I: Bias Originates in the Vision Encoder

The U-MLLM's vision module is built on a vector-quantized variational autoencoder (VQ-VAE) architecture [36]. As illustrated in Figure 6, the encoder compresses an input image into a sequence of discrete latent codes (tokens), which the decoder reconstructs into the original image. The VQ-VAE architecture, upon which the vision module is built, is optimized for high-fidelity image reconstruction. This necessitates the preservation of detailed visual features from the input image, which consequently preserves visually apparent demographic attributes, as their absence would impede the decoder's ability to faithfully render a person's appearance [36]. Thus, as a byproduct of its core objective to enable accurate image reconstruction, the vision tokenizer encodes visually apparent demographic information present in the input images.

**Linear Probing of Image Embeddings**   We conduct a linear probing experiment on the image embeddings. We select a balanced subset of face images from FairFace [19], spanning multiple genders and races, and extract their latent embeddings via the vision encoder. The overall audit pipeline is depicted in Figure 2. Each embedding is paired with its ground-truth demographic label, and we train a linear classifier $\ell(e)$ to predict these labels. As reported in Appendix C, the classifier achieves strong performance: for gender, accuracy of 0.9658, F1 score of 0.9645, recall of 0.9637, and precision of 0.9653; for race (averaged), accuracy of 0.8232, F1 score of 0.8344, recall of 0.8520, and precision of 0.8384. The strong performance of the linear classifier confirms that the vision encoder's latent space retains significant information about demographic attributes from input images, a capability essential for the decoder to achieve high-fidelity reconstruction. It is important to acknowledge that the mere capacity to encode sensitive attributes, while fundamental for reconstruction, does not in itself speak to the module's contribution to fairness [46].

### 3.2 Hypothesis II: Bias Originates from the Language Model

We then investigate whether the language model's token-generation process introduces demographic bias. Let $x_{\text{neutral}}$ denote a neutral prompt (e.g., "a photo of a professor"), and for each demographic attribute $d$ (e.g., "female" or a particular race), let

$$x_{\text{aug}}(d) \;=\; \text{"a photo of a } d \text{ professor".}$$

For each prompt $x \in \{x_{\text{neutral}}\} \cup \{x_{\text{aug}}(d)\}_d$, we use U-MLLM to generate $M$ images, record each image's sequence of discrete image tokens, and predict its demographic label $\hat{d}_i$ via a pre-trained image classifier $C$ [31]. This produces samples

$$\left(x_i,\, y_i,\, \hat{d}_i,\, \mathbf{z}_i\right) \quad \text{for} \quad i = 1, \ldots, M,$$

where $\mathbf{z}_i = (z_{i,1}, z_{i,2}, \ldots)$ is the token sequence for image $y_i$.

**Collecting Image-Token Distributions**    We approximate the model's conditional distribution over token sequences for a prompt $x$ by the empirical distribution

$$\widehat{p}_\theta(\mathbf{z} \mid x) \;=\; \frac{1}{M} \sum_{i=1}^{M} \delta(\mathbf{z} - \mathbf{z}_i) \tag{9}$$

where $\delta(\cdot)$ is a Dirac mass at the observed sequence $\mathbf{z}_i$. In other words, $\widehat{p}_\theta(\mathbf{z} \mid x)$ counts the relative frequency of each generated sequence among $M$ trials. We then measure the Jensen–Shannon divergence (JSD) between the neutral-prompt distribution and each demographic-augmented distribution:

$$D_{\text{JS}}\big(\widehat{p}_\theta(\mathbf{z} \mid x_{\text{neutral}}) \,\big\|\, \widehat{p}_\theta(\mathbf{z} \mid x_{\text{aug}}(d))\big) \tag{10}$$

where for distributions $P$ and $Q$,

$$D_{\text{JS}}(P\|Q) = \tfrac{1}{2} D_{\text{KL}}(P\|M) + \tfrac{1}{2} D_{\text{KL}}(Q\|M), \quad M = \tfrac{1}{2}(P + Q).$$

**Results and Analysis**    For each neutral prompt $x_{\text{neutral}}$, we first determine the majority demographic to which its generated images are assigned. We then identify the augmented-prompt distribution that is closest (in terms of Jensen–Shannon divergence) to the neutral distribution. The hit rate—the proportion of prompts whose neutral distribution aligns most closely with the explicitly specified distribution of its implicit demographic—is $99.80\%$ for gender and $79.34\%$ for race. These results suggest that, when the model implicitly "prefers" a demographic under a neutral prompt (e.g., generating predominantly male images for "a photo of a firefighter"), the token-sequence distribution closely matches that of the corresponding demographic-augmented prompt (e.g., "a photo of a male firefighter"). The high hit rates ($99.80\%$ gender, $79.34\%$ race) indicate that neutral prompts generate image token distributions similar to those from explicitly demographic-augmented prompts reflecting an implicit demographic preference. This suggests that the language model's autoregressive process significantly steers image token selection towards specific groups, with the vision decoder subsequently rendering images.

## 4  Method

### 4.1  Training Data Generation

Obtaining a training dataset with a balanced demographic distribution is challenging. We address this by leveraging a pretrained text-to-image diffusion model, FLUX [20], to synthesize a demographically balanced image set. For each target demographic attribute (e.g., each race–gender combination), we generate images using 1,000 base prompts from training prompts in previous work [34]. For example, given occupation-based prompts such as "a photo of a professor" or "an image of a construction worker," we produce one image per demographic specification (e.g., male, female, Asian, Black) for each prompt. The overall pipeline is illustrated in Figure 8. This process yields a synthetic dataset

$$\mathcal{D}_{\text{bal}} = \big\{(x_j,\, y_j^{(d_1)}, \ldots, y_j^{(d_K)})\big\},$$

where $x_j$ is a neutral prompt (e.g., "a photo of a professor") and $y_j^{(d_i)}$ is the image generated for $x_j$ with demographic attribute $d_i$ (e.g., "a photo of an Asian professor"). By construction, each $x_j$ is paired with exactly one image for each of the $K$ demographic groups, ensuring balanced coverage. Although FLUX [20] is not inherently bias-free, our enumeration strategy enforces demographic variety: naive sampling from neutral prompts resulted in skewed outputs, whereas enforcing one sample per demographic guarantees representation for all groups. In total, we curated 300K images (1000 prompts, 6 demographic groups, 50 images per prompt) to form $\mathcal{D}_{\text{bal}}$. This dataset is used for finetuning (I $\rightarrow$ T) and (T $\rightarrow$ I) as well. Example samples appear in Figure 9.

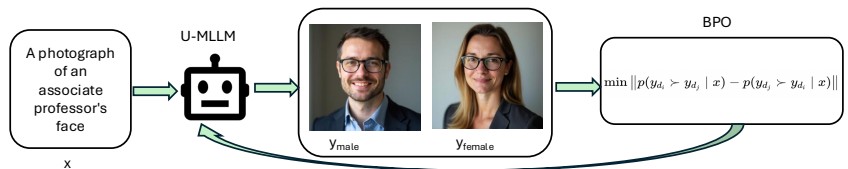

Figure 3: The objective is to minimize deviation of preference between demographic groups.

## 4.2 Balanced Preference Loss

We introduce a balanced preference loss $\mathcal{L}_{\text{bal}}$ to encourage the model to distribute its generation probability evenly across demographics for neutral prompts. Intuitively, we require the model to be indifferent to demographic attributes unless explicitly specified:

$$\min \left| p(y^{(d_i)} \succ y^{(d_j)} \mid x) - p(y^{(d_j)} \succ y^{(d_i)} \mid x) \right| \tag{11}$$

Drawing inspiration from ORPO, we construct $\mathcal{L}_{\text{bal}}$ as a penalty on the pairwise odds ratio between any two demographic groups. For demographic $d_i$ and $d_j$, we define the odds ratio under the model as $\text{OR}_\theta(y^{(d_i)}, y^{(d_j)})$ based on eq. (7), where $y^{(d_i)}$ denotes an image (or image-token sequence) with demographic $d_i$. An $\text{OR}_\theta > 1$ indicates a higher generation likelihood for $d_i$ than for $d_j$. We then define the pairwise balanced preference loss:

$$\mathcal{L}_{\text{bal}}^{(i,j)}(\theta) = \log\left[1 + \left(\sigma(\log \text{OR}_\theta(y^{(d_i)}, y^{(d_j)})) - \tfrac{1}{2}\right)^2\right] \tag{12}$$

where $\sigma$ is the sigmoid function. This term is minimized when $\sigma(\log \text{OR}_\theta) = 0.5$, i.e. $\text{OR}_\theta = 1$ and $p(y^{(d_i)} \mid x) = p(y^{(d_j)} \mid x)$. To extend to $K$ demographics, we sum over all unordered pairs:

$$\mathcal{L}_{\text{bal}}(\theta) = \sum_{1 \leq i < j \leq K} L_{\text{bal}}^{(i,j)}(\theta). \tag{13}$$

Minimizing this aggregate loss encourages a uniform output distribution across demographics.

**Two-Stage Training** We adopt a two-stage training procedure (see Algorithm 1 for details):

1. **Supervised Fine-Tuning (SFT).** We first fine-tune U-MLLM on a supervised dataset of prompt–image-token pairs $\{(x, z)\}$. This stage optimizes the likelihood of the true token sequences, yielding a base model $\pi_{\text{SFT}}$ that produces high-fidelity, semantically accurate images.

2. **Balanced Preference Optimization (BPO).** Starting from $\pi_{\text{SFT}}$, we then minimize the multi-group balanced preference loss $\mathcal{L}_{\text{bal}}$ (via eq. (11)) over the balanced dataset $\mathcal{D}_{\text{bal}}$. This reference-free odds-ratio penalty encourages equal generation preference across all demographics, reducing bias while preserving output quality.

In our implementation, the multi-group loss defined in eq. (13) extends the two-group formulation in eq. (12). We optimize the model parameters $\theta$ to minimize $\mathcal{L}_{\text{bal}}$ after the SFT stage.

**Gradient of the Balanced Preference Loss** The BPO objective admits a compact, closed-form gradient that is both smooth and numerically stable:

$$\nabla_\theta \, \mathcal{L}_{\text{bal}}(\theta) = \underbrace{\frac{2\,(w - 0.5)}{1 + (w - 0.5)^2}}_{\dfrac{\partial \mathcal{L}_{\text{bal}}}{\partial w}} \times \underbrace{w\,(1 - w)}_{\dfrac{\partial w}{\partial v}} \times \underbrace{\left[\frac{\nabla_\theta \, p_\theta(y_{d_i})}{p_\theta(y_{d_i})\,[1 - p_\theta(y_{d_i})]} - \frac{\nabla_\theta \, p_\theta(y_{d_j})}{p_\theta(y_{d_j})\,[1 - p_\theta(y_{d_j})]}\right]}_{\nabla_\theta v(\theta)}$$

where $w = \sigma(v(\theta))$ and $v(\theta) = \log \text{OR}_\theta(y^{(d_i)}, y^{(d_j)})$. When $p_\theta(y^{(d_i)} \mid x) = p_\theta(y^{(d_j)} \mid x)$, we have $w = \tfrac{1}{2}$ and the gradient vanishes, indicating a balanced stationary point. If one demographic's probability is higher, the prefactor flips sign, nudging the model toward equilibrium. A full derivation is provided in Appendix E.

# 5 Experiment

## 5.1 Experimental Setup

**Models and Data.** We evaluate demographic bias (gender and race) in several state-of-the-art U-MLLMs: VILA-U [43], TokenFlow [32], Emu3 [40], Janus [42], Show-o [44], and Janus-Pro [8]. As a baseline, we include diffusion model (Stable Diffusion v1.5). For bias evaluation, we use a set of prompts comprising 50 test examples and 1,000 training examples and adopt metrics from fairness benchmark [34]. Following [34], for each neutral prompt (e.g., occupations or roles without demographic specification), we generate 160 images. Each image is then classified by a pretrained demographic classifier [34] (for gender or race) to compute bias metrics. More details on experimental setup and computational cost can be found in Appendix J. The code can be found at our repository.

**Metrics.** Bias is quantified via the *Relative Diversity* (RD) metric [34], which measures deviation from a uniform demographic distribution:

$$\text{bias}(\mathbf{P}) \;=\; \frac{1}{K(K-1)/2} \sum_{i<j} \big|\text{freq}(i) - \text{freq}(j)\big|,$$

where $\text{freq}(i)$ is the fraction of samples assigned to demographic $d_i$. Lower RD values indicate more balanced representation. We report Gender Bias and Race Bias as the RD over the respective demographic sets, and Intersectional Bias (G×R) over joint gender–race combinations. To assess image quality, we compute the CLIP score [14] (image–text similarity), CLIP-IQA [38] (image quality assessment), and Inception Score (image diversity) [5]. These metrics verify that debiasing does not compromise output fidelity or relevance.

## 5.2 Experimental Results

Table 1: Image generation bias.

| Debias: | Method | Bias ↓ | | | Image Quality ↑ | | |
|---|---|---|---|---|---|---|---|
| | | Gender | Race | G.×R. | CLIP-S | CLIP-IQA | Inception |
| | Stable Diffusion | 0.66 | 0.41 | 0.21 | 27.69 | 0.67 | — |
| | Janus | 0.87 | 0.43 | 0.23 | 27.44 | 0.69 | 2.27 |
| | Janus-Pro | 0.90 | 0.48 | 0.24 | 27.62 | 0.82 | 1.79 |
| | Show-o | 0.85 | 0.48 | 0.24 | 27.16 | 0.86 | 1.79 |
| | TokenFlow | 0.84 | 0.47 | 0.24 | 27.17 | 0.84 | 2.34 |
| | VILA-U | 0.89 | 0.48 | 0.24 | 28.24 | 0.84 | 1.87 |
| Gender | Prompt Engineering | 0.56 | 0.49 | 0.23 | 28.51 | 0.82 | 1.91 |
| | Fine-tune(I → T) | 0.83 | 0.42 | 0.22 | 28.49 | 0.80 | 2.28 |
| | Fine-tune(T → I) | 0.27 | 0.51 | 0.23 | 27.66 | 0.77 | 1.85 |
| | BPO | **0.25** | 0.50 | 0.22 | 27.74 | 0.77 | 2.10 |
| Race | Prompt Engineering | 0.56 | 0.49 | 0.23 | 28.51 | 0.82 | 1.91 |
| | Fine-tune(I → T) | 0.78 | 0.44 | 0.22 | 28.14 | 0.80 | 2.54 |
| | Fine-tune(T → I) | 0.83 | **0.23** | 0.17 | 27.98 | 0.80 | 1.98 |
| | BPO | 0.78 | 0.26 | 0.18 | 27.66 | 0.81 | 2.31 |
| G.×R. | Prompt Engineering | 0.59 | 0.33 | 0.18 | 28.09 | 0.80 | 1.87 |
| | Fine-tune(I → T) | 0.86 | 0.45 | 0.23 | 28.34 | 0.82 | 2.22 |
| | Fine-tune(T → I) | 0.46 | 0.32 | 0.17 | 27.90 | 0.78 | 1.91 |
| | BPO | 0.52 | 0.26 | **0.15** | 27.78 | 0.80 | 2.06 |

**Bias in Baseline Models.** Table 1 presents bias and quality metrics for Stable Diffusion(SD) and U-MLLMs. All baselines exhibit *substantial* demographic bias: gender scores range from 0.66 (SD) to 0.90 (Janus-Pro), while race scores fall between 0.41 and 0.48. VILA-U is the most gender-skewed baseline (0.89), corroborating our earlier qualitative findings. Image quality is uniformly high (CLIP-S ≈27–28), with Show-o attaining the best perceptual quality (CLIP-IQA 0.86) albeit with pronounced bias. These results confirm that contemporary U-MLLMs generate visually appealing images yet systematically favor particular demographics, underscoring the need for targeted debiasing.

**Effect of Our Debiasing Method.** We fine-tune VILA-U with our two-stage *Balanced Preference Optimization* (BPO) in three settings: *BPO-Gender*, *BPO-Race*, and *BPO-Mix* (joint debiasing). On

the gender setting, **BPO-Gender** reduces bias to **0.25**—a $\sim 72\%$ drop relative to the baseline—while preserving CLIP-S (27.74) and yielding a strong Inception score (2.10). For race, **Fine-tune (T$\rightarrow$ I)** gives the lowest score (0.23), but **BPO-Race** is competitive (0.26) with a higher Inception score (2.31). When debiasing both axes simultaneously, **BPO-Mix** achieves the best joint score of **0.15**, outperforming prompt engineering (0.18) and fine-tuning (0.17) while maintaining image quality (Inception 2.06, CLIP-IQA 0.80). BPO consistently delivers the strongest bias reduction—especially for joint gender–race bias—without sacrificing fidelity, making it a practical debiasing strategy.

**Generalization.** **(1) Stereotype Prompts [45].** Test results on the 564 stereotype prompts are presented in Table 4, *BPO–Gender* reduces VILA-U's gender-bias score from 0.6467 to 0.4086 ($\sim 37\%$ drop). *BPO–Race* lowers race bias from 0.4310 to 0.2566 ($\sim 40\%$ reduction). Applying *BPO–Mix* (joint debiasing) yields gender and race scores of 0.4629 and 0.2880, and cuts intersectional bias from 0.2161 to 0.1622. Across all settings, CLIP-S and CLIP-IQA remain largely intact, indicating that debiasing does not degrade image quality. **(2) Cross-Language Generalization.** To probe robustness across languages, we translated the evaluation prompts of [34] into Chinese and French using the ChatGPT API. As summarized in Table 5, the baseline gender-bias scores are 0.4789 (Chinese) and 0.8542 (French). After *BPO–Gender*, these drop to 0.2810 and 0.3000, respectively—mirroring the reductions observed in English. CLIP-S and CLIP-IQA stay on par with the English case, demonstrating that our balanced loss effectively mitigates demographic bias *across* datasets and languages while preserving image fidelity.

## 5.3 Discussion

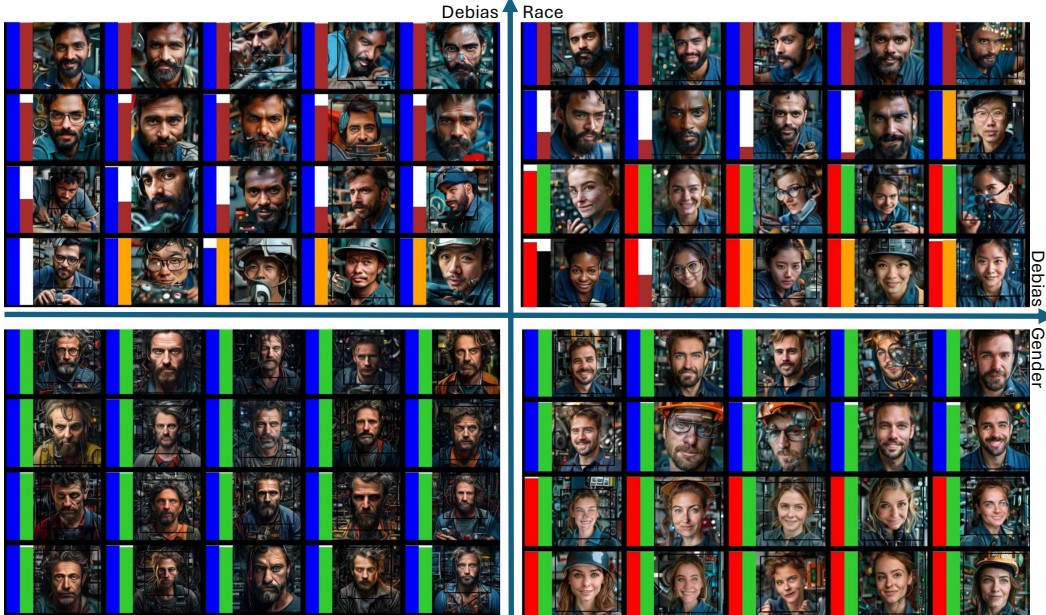

Figure 4: Given the test prompt: "a photo of the face of an electrical and electronics repairer, a person". Images generated by: VILA-U (bottom-left), our method reduces bias in gender (bottom-right), race (top-left), gender-race intersection (top-right). For each image, the first color-coded bar represents the predicted gender: male or female. The second bar represents the predicted race: White, Asian, Black, or Indian. The height of each bar indicates the prediction confidence.

**Qualitative Samples** Figure 4 shows VILA-U's outputs with and without our debiasing method for a representative prompt. The qualitative examples demonstrate that our approach enhances demographic diversity without degrading visual quality. For instance, given the prompt "a photo of the face of an electrical and electronics repairer, a person" (which does not specify any demographic), the original VILA-U predominantly generates images of men of a single race, whereas the debiased model produces a varied mix of genders and races. More samples are in Figure 11.

**Understanding vs. Generative Bias** Beyond visual inspection, we evaluate whether bias arises in the model's internal understanding. To assess bias in understanding tasks, we run each model and

sample multiple times in a pure VQA setting with an empty image, asking, "What is the gender of occupation? (male/female/unknown)." This removes visual cues and reveals inherent perceptual bias. As shown in Figure 1, most models still show bias to some extent. Some models, such as Janus, however, often reply "unknown" or refuse to specify, indicating reduced explicit bias due to alignment training. Despite this, their generative outputs remain skewed (e.g., Janus still produces images heavily favoring one demographic). This partial alignment—neutral Q&A behavior but biased generation—highlights that aligning the model's textual responses does not suffice to eliminate bias in visual generation.

**Can debiasing understanding help debias generation?** Prior work [35] suggests that improving image understanding may indirectly benefit generation. Inspired by this, we fine-tuned U-MLLM in image-to-text to reduce its understanding bias (see Figure 10), yielding the fine-tune (I→T) variant. However, as shown in Table 1, this approach yields only marginal reductions in generation bias (Gender: 0.89→0.83; Race: 0.48→0.44; Intersectional: 0.24→0.23). Thus, teaching the model that different races/genders share an occupation does not alter its generative bias. These findings imply that bias in *understanding* and bias in *generation* can arise from distinct mechanisms. Consequently, improving one aspect of visual–linguistic alignment does not guarantee fairness in the other. Future methods must jointly address both *visual comprehension* and *visual generation* biases. Additionally, prior work [6] identified the "Reversal Curse": LLMs trained on "A is B" often fail to learn the inverse "B is A.". This phenomenon mirrors our finding that, even after teaching the U-MLLM to map both "male truck driver" and "female truck driver" to the same concept "truck driver," the model continues to exhibit bias when given a neutral prompt(see Figure 10).

**Additional Loss Variants** In addition to our core BPO objective, we experimented with several alternative loss formulations. For instance, we tested a thresholded penalty that activates only when the probability of one demographic exceeds another by a fixed margin, as well as a simpler loss on log-probability differences (omitting the $(1 - p)$ terms). The threshold-based variants required careful margin tuning and caused training instability, while the log-probability difference loss was prone to gradient explosion when probabilities approached 0 or 1. For full definitions and empirical comparisons, see Appendix F.

# 6 Related Work

**Multimodal Generative Models** Unified multimodal large language models (U-MLLMs) have advanced the state-of-the-art by bridging visual understanding and conditional generation capabilities. Compared to early MLLMs, which focus purely on understanding, such as LlaVA series [27, 26], these more recent works, represented by VILA-U [43], Show-o [44], MetaMorph [35], TokenFlow [32], Emu3 [40], TokenFusion [47], Janus [42, 28], etc. [3, 21, 25], highlight their effectiveness in generating high-quality visuals conditioned on text prompts. U-MLLMs usually employ autoregressive paradigms that may inherit or amplify the biases embedded in their training data. Existing studies predominantly focus on performance improvement rather than understanding and addressing the more critical issues such as demographic fairness.

**Fairness in Image Generation** The social risks of biased image generation, particularly gender and racial disparities, have been extensively documented [18, 22, 23]. Prior efforts in diffusion-based models attempted to mitigate bias by re-balancing training data and incorporating fairness objectives [17]. However, these approaches are not directly transferable to U-MLLMs due to architectural differences and tokenization mechanisms. Few studies explore bias sources within unified models or their downstream effects on generated outputs, leaving a gap in understanding the interaction between text and image modalities.

**Preference Optimization** Direct preference optimization (DPO) [33] has emerged as a promising technique to address biases in machine learning models, especially LLM. Since then, numerous new loss functions have been proposed [29, 31, 15, 12, 2]. Recent advances [1] integrate preference modeling into loss functions to guide models toward balanced outputs, which have rarely been explored for MLLMs. Based on this, we introduce a novel balanced preference loss tailored for U-MLLMs. By leveraging demographic attributes during training, the proposed method balances the likelihood of generating outputs across groups without compromising the image quality.

# 7   Conclusion

We examined demographic bias (gender and race) in U-MLLMs for image generation and proposed a method to mitigate it. Our study found that current U-MLLMs often produce images with skewed demographics when given neutral prompts. Through our component-wise analysis, the findings suggest that the language model within the U-MLLM is a primary driver of the observed demographic bias in generated images when given neutral prompts. To address the issue, we introduced a two-stage debiasing approach: first fine-tuning the model on a balanced dataset, and then applying a balanced preference loss inspired by direct preference optimization. This approach significantly reduced the bias in generated images across gender and race categories while preserving image quality and relevance. We also discovered a partial alignment phenomenon where models might appear unbiased in a textual response yet remain biased in visual generation, underscoring the need for generation-specific debiasing techniques. Our work provides an initial framework for auditing and improving the fairness of U-MLLMs. We hope it encourages the development of more holistic debiasing strategies that consider all aspects of multimodal model behavior. In the future, we plan to extend our method to additional demographic dimensions (e.g., age, ethnicity). Ensuring fairness in generative models is critical as these models become increasingly influential in content creation. Our work does have some limitation as illustrated in Appendix A.

# 8   Acknowledgments

This paper is partially supported by the Commonwealth Cyber Initiative (CCI) program H-2Q25-020. The authors acknowledge the Modal Academic Compute Credit and William & Mary Research Computing for providing computational resources and/or technical support that have contributed to the results reported within this paper. URL: https://www.wm.edu/it/rc.

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

## A  Limitation

**Limitation.** Several questions remain open despite the bias reduction achieved by our approach. First, our study primarily centered on *overt* demographic categories (e.g., gender, race). Real-world scenarios may demand addressing *intersectional* or *nuanced* attributes (e.g., age, culture, or religion). Second, many models are not fully open-source, restricting the scope of our evaluations to publicly available systems. Future research could broaden the range of tested models. Third, our definition of fairness as demographic parity and our use of discrete, Western-centric gender and racial categories are specific value judgments that may not apply universally and risk oversimplifying complex identities. We also recognize that our own perspectives as researchers have shaped the project's scope, our method defines fairness as demographic parity, but this is just one interpretation. Lastly, due to resource constraints, we did not explore alternative preference optimization objectives beyond our framework. Building on our method to incorporate other debiasing approaches is a promising direction for future work.

## B  Structure of U-MLLM

This section presents the structure of VILA-U and its visual tokenizer; all the content in this section is from prior study [43].

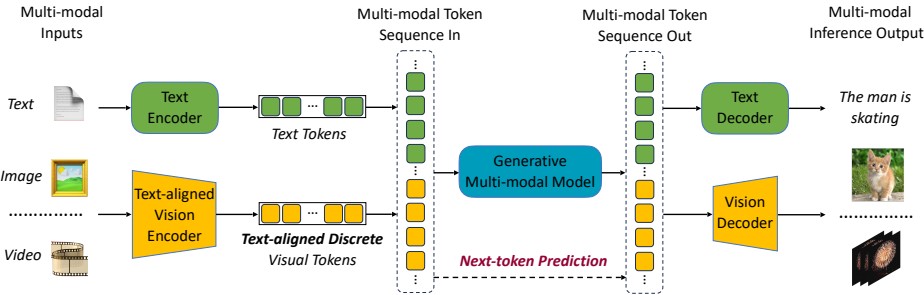

Figure 5: **Overview of framework's multi-modal training and inference process[43]** Visual inputs are converted into discrete tokens and merged with textual tokens to create a unified multi-modal token sequence. This sequence is used in next-token prediction process, which supports a unified training objective. During inference, output tokens are processed through either text detokenizer or vision tower decoder, generating multi-modal content outputs[43].

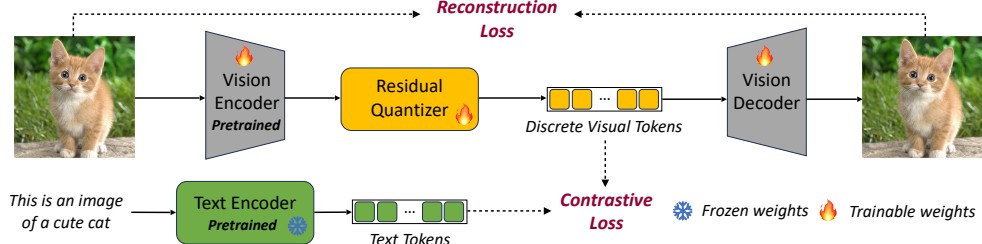

Figure 6: **Overview of unified foundation vision tower[43]** Input images are processed by the vision encoder, where features are extracted and discretized using residual quantization. These discrete vision features are then utilized in two ways: they are fed into the vision decoder to reconstruct images and are used to perform text-image alignment. Throughout this process, both the reconstruction loss and contrastive loss are calculated to refine the vision tower, enabling it to generate discrete visual features that are aligned with text[43].

# C  Locating Bias

|  | Gender | Race |
|---|---|---|
| **Hit Rate** | 0.9980 | 0.7934 |

Table 2: JS-Divergence Hit Rates for Gender and Race Variant Prompts. For each training prompt, images were generated using both a neutral prompt and prompt variants in gender or race. We first determined the majority gender/race from the neutral-prompt images (skipping ambiguous cases). We then counted a "hit" whenever the JS divergence between the neutral prompt's image-token distribution and the matching-variant prompt's distribution was smaller than that for non-matching variants. More than 99% of prompts yield closer divergence for gender-matching variants, and about 79% for race-matching variants, indicating that the language model's token sampling is skewed toward certain demographics.

| Pair | Accuracy | F1 | Recall | Precision |
|---|---|---|---|---|
| Male vs. Female | 0.9658 | 0.9645 | 0.9637 | 0.9653 |
| Black vs. Southeast Asian | 0.9167 | 0.9187 | 0.9658 | 0.8760 |
| Black vs. Indian | 0.8958 | 0.8988 | 0.9328 | 0.8672 |
| Black vs. Middle Eastern | 0.4667 | 0.6364 | 1.0000 | 0.4667 |
| Black vs. East Asian | 0.9500 | 0.9492 | 0.9739 | 0.9256 |
| Black vs. Latino Hispanic | 0.8333 | 0.8113 | 0.7107 | 0.9451 |
| Black vs. White | 0.9167 | 0.9231 | 0.9091 | 0.9375 |
| Southeast Asian vs. Indian | 0.9208 | 0.9156 | 0.8879 | 0.9450 |
| Southeast Asian vs. Middle Eastern | 0.9333 | 0.9316 | 0.8934 | 0.9732 |
| Southeast Asian vs. East Asian | 0.6458 | 0.6222 | 0.5385 | 0.7368 |
| Southeast Asian vs. Latino Hispanic | 0.8625 | 0.8546 | 0.8509 | 0.8584 |
| Southeast Asian vs. White | 0.9042 | 0.9013 | 0.8750 | 0.9292 |
| Indian vs. Middle Eastern | 0.8958 | 0.8980 | 0.8661 | 0.9322 |
| Indian vs. East Asian | 0.9417 | 0.9381 | 0.9381 | 0.9381 |
| Indian vs. Latino Hispanic | 0.7625 | 0.7595 | 0.7563 | 0.7627 |
| Indian vs. White | 0.8917 | 0.8992 | 0.8923 | 0.9062 |
| Middle Eastern vs. East Asian | 0.5208 | 0.6849 | 1.0000 | 0.5208 |
| Middle Eastern vs. Latino Hispanic | 0.6875 | 0.6445 | 0.5862 | 0.7158 |
| Middle Eastern vs. White | 0.7458 | 0.7359 | 0.7083 | 0.7658 |
| East Asian vs. Latino Hispanic | 0.9167 | 0.9180 | 0.9655 | 0.8750 |
| East Asian vs. White | 0.9333 | 0.9322 | 0.9402 | 0.9244 |
| Latino Hispanic vs. White | 0.7458 | 0.7490 | 0.7000 | 0.8053 |

Table 3: Embedding classification metrics by pairwise comparisons.

# D BPO Algorithm

---

**Algorithm 1** Balanced Preference Optimization

---

**Input**: U-MLLM with parameters $\theta_0$; SFT dataset $\mathcal{D}_{\text{SFT}} = \{(x_i, z_i)\}$ of prompts $x_i$ and image tokens $z_i$; Balanced dataset $\mathcal{D}_{\text{bal}} = \{(x_j, y_{j_1}, \ldots, y_{j_K})\}$, each $x_j$ with multiple demographic variants; Trade-off parameter $\lambda$, total training epochs $N_1, N_2$; **Output**: Debiased model parameters $\theta$

1: **Stage 1: Supervised Finetuning**
2:     Initialize $\theta \leftarrow \theta_0$
3: **for** epoch $= 1$ to $N_1$ **do**
4:     Sample a minibatch $\{(x_i, z_i)\}$ from $\mathcal{D}_{\text{SFT}}$
5:     $\mathcal{L}_{\text{NLL}}(\theta) = -\sum_{(x_i, z_i)} \log P_\theta(z_i \mid x_i)$
6:     Update $\theta \leftarrow \theta - \eta \nabla_\theta \mathcal{L}_{\text{NLL}}(\theta)$
7: **end for**
8: **Stage 2: Balanced Preference Optimization**
9: **for** epoch $= 1$ to $N_2$ **do**
10:     Sample a minibatch $\{(x_j, y_{j_1}, \ldots, y_{j_K})\}$ from $\mathcal{D}_{\text{bal}}$
11:     **Balanced Preference Loss:**

$$\mathcal{L}_{\text{bal}}(\theta) = \sum_{k \neq l} \mathcal{L}_{\text{bal}}^{(d_{k-1}, d_k)}(\theta).$$

12:     Update $\theta \leftarrow \theta - \eta \nabla_\theta \mathcal{L}_{\text{bal}}(\theta)$
13: **end for**
14: **Return** $\theta$

---

# E  Gradient of BPO

## E.1  Setup: The Balanced Preference Loss

For simplicity, consider just two demographic categories $d_i$ and $d_j$. The extension to multiple groups is done by summing over pairs.

We define the *odds ratio*:

$$\mathrm{OR}_\theta(y_{d_i}, y_{d_j}) \;=\; \frac{\mathrm{odds}_\theta(y_{d_i})}{\mathrm{odds}_\theta(y_{d_j})}, \quad \text{where} \quad \mathrm{odds}_\theta(y_d) \;=\; \frac{p_\theta(y_d \mid x)}{1 - p_\theta(y_d \mid x)}.$$

Taking the logarithm,

$$v(\theta) \;=\; \log \mathrm{OR}_\theta(y_{d_i}, y_{d_j}) \;=\; \underbrace{\log \mathrm{odds}_\theta(y_{d_i})}_{\log\left[p_\theta(y_{d_i}|x)\right] \,-\, \log\left[1-p_\theta(y_{d_i}|x)\right]} \;-\; \underbrace{\log \mathrm{odds}_\theta(y_{d_j})}_{\log\left[p_\theta(y_{d_j}|x)\right] \,-\, \log\left[1-p_\theta(y_{d_j}|x)\right]} \; .$$

Next, let $w(\theta) = \sigma\big(v(\theta)\big)$, where $\sigma$ is the logistic sigmoid $\sigma(z) = \frac{1}{1+e^{-z}}$.

The **Balanced Preference Loss** for two groups is:

$$\mathcal{L}_{\mathrm{bal}}(\theta) = \log\!\Big[1 + \big(\underbrace{w(\theta) - \tfrac{1}{2}}_{\sigma(v)}\big)^{2}\Big].$$

Minimizing this loss pushes $w(\theta) \to \frac{1}{2}$, i.e. $\mathrm{OR}_\theta \to 1$, which implies $p_\theta(y_{d_i} \mid x) = p_\theta(y_{d_j} \mid x)$.

## E.2  Chain Rule for $\nabla_\theta \mathcal{L}_{\mathbf{bal}}$

We want:

$$\nabla_\theta \mathcal{L}_{\mathrm{bal}}(\theta) \;=\; \frac{\partial \mathcal{L}_{\mathrm{bal}}}{\partial w} \,\times\, \frac{\partial w}{\partial v} \,\times\, \nabla_\theta v(\theta).$$

We handle each factor separately.

**Derivative of $\mathcal{L}_{\mathbf{bal}}$ w.r.t. $w$**

Set

$$r(\theta) \;=\; w(\theta) - \tfrac{1}{2} \;=\; \sigma\big(v(\theta)\big) - \tfrac{1}{2}.$$

Then

$$\mathcal{L}_{\mathrm{bal}}(\theta) \;=\; \log\!\big[1 + (r(\theta))^{2}\big].$$

Hence,

$$\frac{\partial \mathcal{L}_{\mathrm{bal}}}{\partial w} \;=\; \frac{\partial \mathcal{L}_{\mathrm{bal}}}{\partial r} \,\times\, \frac{\partial r}{\partial w} \;=\; \frac{1}{1 + r^{2}}\big[2\,r\big] \,\times\, 1 \;=\; \frac{2\,r}{1 + r^{2}} \;=\; \frac{2\,(w - 0.5)}{1 + (w - 0.5)^{2}}.$$

**Derivative of $w = \sigma(v)$ w.r.t. $v$**

We know $\frac{d}{dv}\sigma(v) = \sigma(v)\big(1 - \sigma(v)\big)$. So

$$\frac{\partial w}{\partial v} \;=\; w(\theta)\big[1 - w(\theta)\big] \;=\; \sigma\big(v(\theta)\big)\big[1 - \sigma\big(v(\theta)\big)\big].$$

**Gradient of $v(\theta)$ w.r.t. $\theta$**

Recall

$$v(\theta) \;=\; \log \mathrm{OR}_\theta(y_{d_i}, y_{d_j}) \;=\; \log \mathrm{odds}_\theta(y_{d_i}) \;-\; \log \mathrm{odds}_\theta(y_{d_j}).$$

Hence,

$$\nabla_\theta v(\theta) \;=\; \nabla_\theta\Big[\log \mathrm{odds}_\theta(y_{d_i})\Big] \;-\; \nabla_\theta\Big[\log \mathrm{odds}_\theta(y_{d_j})\Big].$$

We have
$$\log \text{odds}_\theta(y_d) = \log p_\theta(y_d \mid x) - \log\big[1 - p_\theta(y_d \mid x)\big].$$

Thus,
$$\nabla_\theta \log \text{odds}_\theta(y_d) = \frac{1}{p_\theta(y_d \mid x)} \nabla_\theta p_\theta(y_d \mid x) + \frac{1}{1 - p_\theta(y_d \mid x)} \nabla_\theta p_\theta(y_d \mid x),$$

where we used $\nabla_\theta \log z(\theta) = \frac{1}{z}\nabla_\theta z$. Combine the two terms carefully (and noting the minus sign in the second log derivative switches sign again), we get:

$$\nabla_\theta \log \text{odds}_\theta(y_d) = \frac{1}{p_\theta(y_d \mid x)\big[1 - p_\theta(y_d \mid x)\big]} \nabla_\theta p_\theta(y_d \mid x).$$

Putting it all together for $d_i$ and $d_j$:

$$\nabla_\theta v(\theta) = \frac{1}{p_\theta(y_{d_i})\big[1 - p_\theta(y_{d_i})\big]} \nabla_\theta p_\theta(y_{d_i}) - \frac{1}{p_\theta(y_{d_j})\big[1 - p_\theta(y_{d_j})\big]} \nabla_\theta p_\theta(y_{d_j}).$$

(Here we have suppressed the conditioning on $x$ in notation, just to keep it lighter.)

### E.3 Final Gradient Expression

By combining previous sections, we get:

$$\boxed{\nabla_\theta \, \mathcal{L}_{\text{bal}}(\theta) = \underbrace{\frac{2\,(w - 0.5)}{1 + (w - 0.5)^2}}_{\frac{\partial \mathcal{L}_{\text{bal}}}{\partial w}} \times \underbrace{w\,(1 - w)}_{\frac{\partial w}{\partial v}} \times \underbrace{\left[ \frac{\nabla_\theta \, p_\theta(y_{d_i})}{p_\theta(y_{d_i})\big[1 - p_\theta(y_{d_i})\big]} - \frac{\nabla_\theta \, p_\theta(y_{d_j})}{p_\theta(y_{d_j})\big[1 - p_\theta(y_{d_j})\big]} \right]}_{\nabla_\theta v(\theta)}}.$$

where $w(\theta) = \sigma\big(v(\theta)\big)$.

Interpretation:

- If $p_\theta(y_{d_i}) \gg p_\theta(y_{d_j})$, the odds ratio is large, so $v(\theta)$ is large and $w(\theta) \approx 1$. The factor $(w - 0.5)$ is then positive and big, so the gradient pushes parameters $\theta$ to reduce $p_\theta(y_{d_i})$ (and/or raise $p_\theta(y_{d_j})$).

- Conversely, if $p_\theta(y_{d_i}) \ll p_\theta(y_{d_j})$, we get a negative factor in front of $\nabla_\theta p_\theta(y_{d_i})$.

- When $p_\theta(y_{d_i}) = p_\theta(y_{d_j})$, then $v(\theta) = 0$ and $w(\theta) = 0.5$. The entire derivative is zero, which is precisely the balanced solution we want.

### E.4 Multiple Demographic Groups

For more than two groups, say $\{d_1, \ldots, d_K\}$, one can (as in the paper) sum or average pairwise losses:
$$\mathcal{L}_{\text{bal}}^{(\text{multi})}(\theta) = \sum_{1 \le i < j \le K} \mathcal{L}_{\text{bal}}(\theta; d_i, d_j).$$

Then $\nabla_\theta \mathcal{L}_{\text{bal}}^{(\text{multi})}$ is just the sum of the two-group gradients, ensuring all pairwise distributions converge to balance.

### E.5 Putting It All

This derivation shows exactly **how** to compute the gradient of the Balanced Preference Loss. It also clarifies **why** the only way to get a zero-gradient solution for each pair $(d_i, d_j)$ is to **equalize** their probabilities $p_\theta(y_{d_i} \mid x) = p_\theta(y_{d_j} \mid x)$. Hence the uniform distribution across demographics is the unique global optimum (apart from mild edge cases).

# F Comparison with Other Loss Functions

In our experiments, we evaluated several loss formulations designed to reduce demographic bias. Below we describe four variants of odds-ratio penalties, each designed to encourage balanced treatment between pair of demographic groups $d_i$ and $d_j$. For brevity, we set

$$p_{d_i} = p_\theta(y_{d_i} \mid x) \quad \text{and} \quad p_{d_j} = p_\theta(y_{d_j} \mid x).$$

For the methods below, we also define the log-odds difference as

$$\Delta = \log(p_{d_i}) - \log(p_{d_j}).$$

## F.1 Method 1 (Our Core Approach)

**Key Idea:** Penalize large deviations in the log-odds difference $\Delta$ (Figure 7). Specifically, we define

$$\Delta = \log\left(\frac{p_{d_i}}{1 - p_{d_i}}\right) - \log\left(\frac{p_{d_j}}{1 - p_{d_j}}\right)$$

and then compute the balanced loss as

$$\ell_{\text{bal}}(\Delta) = \log\left(1 + \left[\sigma(\Delta) - 0.5\right]^2\right),$$

where $\sigma(\Delta) = \frac{1}{1+e^{-\Delta}}$ is the sigmoid function. When $\sigma(\Delta) \approx 0.5$, the model is equally likely to generate $y_{d_i}$ or $y_{d_j}$, so $\ell_{\text{bal}}$ is near 0. Larger imbalances yield a smoothly increasing penalty. The overall loss is obtained by averaging $\ell_{\text{bal}}$ over pair $\{(d_i, d_j)\}$.

**Why It Works Best:**

- It employs the true log-odds, $\log\left(\frac{p}{1-p}\right)$, which remains stable even when $p$ is near 0 or 1.
- The function $\ell_{\text{bal}}(\Delta)$ is continuously differentiable for all $\Delta$, avoiding the discontinuities that arise in thresholded formulations.
- In practice, Method 1 produces stable training dynamics and superior bias reduction.

## F.2 Method 2: Thresholded Loss on Log Probability Ratios

A simpler approach uses the ratio of probabilities directly. Define

$$\delta = \sigma\left(\log\left(\frac{p_{d_i}}{p_{d_j}}\right)\right) - 0.5.$$

A piecewise penalty is then applied only when $|\delta|$ exceeds a fixed threshold $t$:

$$\ell_{\text{thresh}}(\delta) = \begin{cases} \alpha \left|\delta - t\right|, & \text{if } |\delta| \geq t, \\ 0, & \text{otherwise}, \end{cases}$$

where $\alpha$ is a penalty scale.

**Drawbacks:**

- It requires choosing both a threshold $t$ and a scale $\alpha$, which may be sensitive to the task at hand.
- The training process can become unstable near the threshold boundaries.

## F.3 Method 3: Thresholding on True Log-Odds

This variant is similar to Method 2 but uses the full log-odds difference. With

$$\Delta = \log(p_{d_i}) - \log(p_{d_j}),$$

a piecewise penalty is applied if the corresponding sigmoid value deviates from 0.5 beyond a threshold:

$$\ell_{\text{thresh}}(\Delta) = \begin{cases} \alpha \left|\sigma(\Delta) - 0.5 - t\right|, & \text{if } \left|\sigma(\Delta) - 0.5\right| \geq t, \\ 0, & \text{otherwise}. \end{cases}$$

It still suffers from the need to tune the threshold $t$ and stabilization.

### F.4 Method 4: Plain Log Probability Difference Squared

A straightforward approach computes the difference in log probabilities:

$$\Delta_{\text{simple}} \;=\; \log\!\big(p_{d_i}\big) - \log\!\big(p_{d_j}\big)$$

and then defines the loss as

$$\ell_{\text{sq}}(\Delta_{\text{simple}}) \;=\; \left[\frac{1}{2} - \sigma\!\big(\Delta_{\text{simple}}\big)\right]^2.$$

Because this method does not incorporate the $1 - p$ terms found in the log, it is susceptible to numerical instability when $p_{d_i}$ or $p_{d_j}$ approaches 0 or 1. Empirically, it tends to perform slightly worse than Method 1.

### F.5 Empirical Observations

The training loss for each method can be found in Figure 7. **Method 1** consistently achieves the best balance between fairness and training stability. **Threshold-based approaches** (Methods 2 and 3) may reduce bias but require careful tuning of the threshold $t$, which may lead to less smooth training. **Method 4** works in moderate probability ranges but is prone to exploding or vanishing gradients when probabilities are near the boundaries of 0 or 1. In our ablation studies (Figure 7), Method 1 outperformed the other variants by achieving lower bias metrics along with stable training dynamics and minimal computational overhead. Based on these observations, we selected Method 1 as our primary Balanced Preference Optimization (BPO) objective.

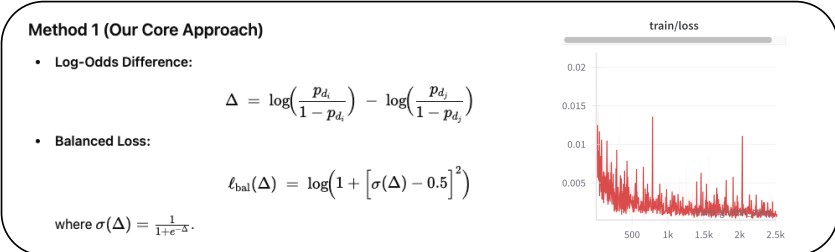

(a) Training Loss plot for method 1(core method).

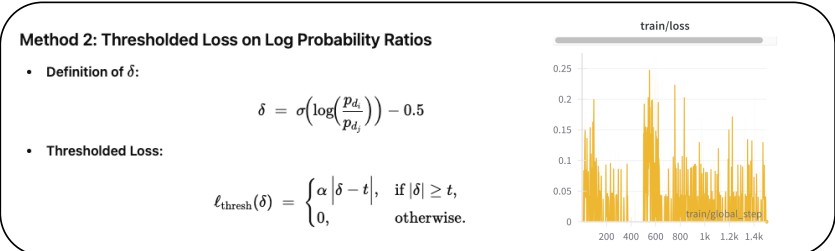

(b) Training Loss plot for method 2.

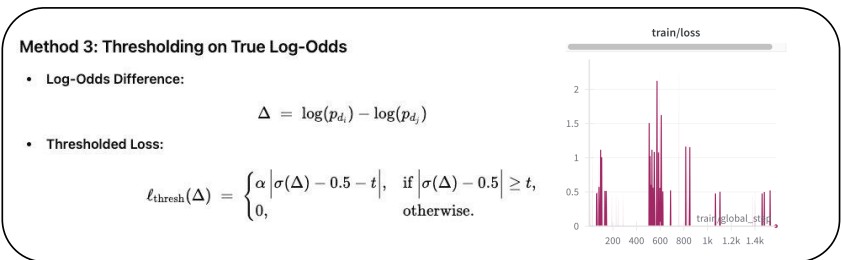

(c) Training Loss plot for method 3.

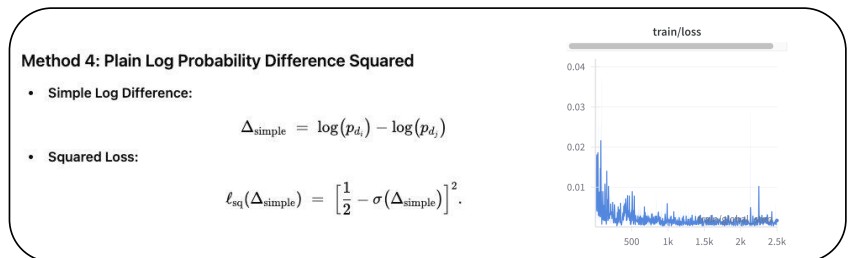

(d) Training Loss plot for method 4.

Figure 7: We define $p_{d_i} = p_\theta(y_{d_i} \mid x)$ and $p_{d_j} = p_\theta(y_{d_j} \mid x)$. We compare four loss formulations: (1) Method 1 penalizes deviations in the log-odds difference $\Delta = \log\left(\frac{p_{d_i}}{1-p_{d_i}}\right) - \log\left(\frac{p_{d_j}}{1-p_{d_j}}\right)$ using a continuously differentiable loss $\ell_{\text{bal}}(\Delta) = \log\left(1 + [\sigma(\Delta) - 0.5]^2\right)$, yielding stable training and superior bias reduction. (2) Methods 2 and 3 apply thresholded penalties on the log probability ratio or log-odds, requiring careful tuning of a threshold $t$ and potentially causing instability. (3) Method 4 uses the plain squared difference of log probabilities, but can suffer from numerical instability when probabilities approach 0 or 1. Among these, Method 1, Balanced Preference Optimization (BPO), stabilizes the loss and yields the best final results.

# G   Generalization

Table 4: Results on 564 Stereotype Prompts (MMDecodingTrust)

| Method | Gender Bias | Race Bias | Intersection Bias | Avg. CLIP Score | Avg. CLIP IQA Score |
|---|---|---|---|---|---|
| VILA-U | 0.6467 | 0.4310 | 0.2161 | 28.2198 | 0.8577 |
| BPO-gender | 0.4086 | 0.4899 | 0.2256 | 25.1070 | 0.7072 |
| BPO-race | 0.6313 | 0.2566 | 0.1659 | 24.3783 | 0.7466 |
| BPO-mix | 0.4629 | 0.2880 | 0.1622 | 24.5264 | 0.7481 |

Table 5: Preliminary Cross-lingual Test Results (Chinese and French)

| Setting | Gender Bias | Intersection Bias | CLIP Score | CLIP IQA Score |
|---|---|---|---|---|
| VILA-U (CN) | 0.4789 | 0.1938 | 24.3673 | 0.7445 |
| VILA-U (FR) | 0.8542 | 0.2381 | 26.9307 | 0.8853 |
| BPO-gender (CN) | 0.2810 | 0.1983 | 23.8048 | 0.7046 |
| BPO-gender (FR) | 0.3000 | 0.2275 | 25.6598 | 0.7768 |

# H Data Generation Pipeline and Samples

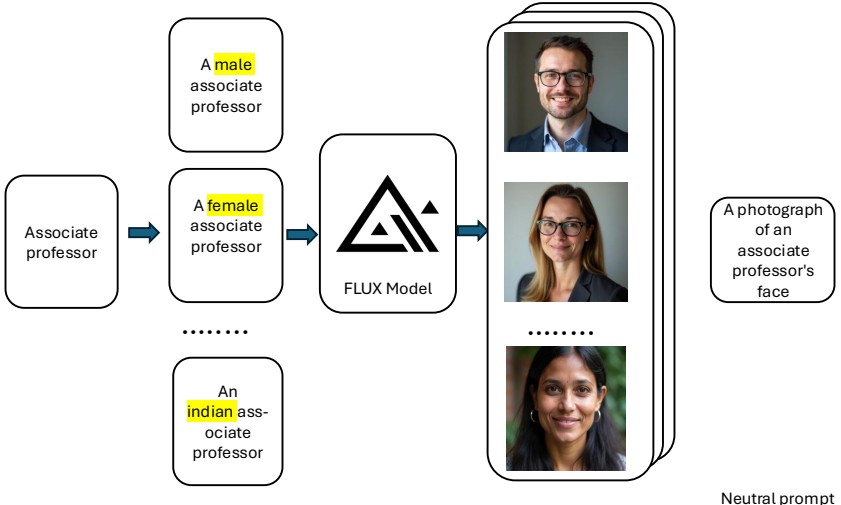

Figure 8: Pipeline for data generation. **Augmented Prompts for Each Demographic:** We explicitly instruct the model with prompts like "Indian associate professor" rather than relying on FLUX.1-dev's default distribution. Each occupation is re-generated for every demographic, and direct instructions yield images with the desired attributes; **Balancing Across All Attributes:** After collecting images for each demographic prompt, we pair them with a neutral prompt (e.g., "a photo of an associate professor"). This creates a training set where each neutral prompt has one sample per demographic, ensuring uniform coverage across races and genders. As a result, the model learns that the same description (e.g., "associate professor") can represent multiple demographic groups, countering one-sided associations; **Goal: A Balanced Synthetic Dataset from a potentially biased Generator:** Our method ensures equal representation per prompt by explicitly enumerating each demographic label and quality-checking outputs. FLUX.1-dev's biases are corrected, as we only require it to produce sufficient images for each category. Thus, while we concur that FLUX.1-dev can exhibit low diversity in unconstrained settings, our augmented-prompt + balanced-pairing pipeline specifically mitigates those limitations and yields a controlled, demographically balanced dataset that is well-suited for our debiasing objectives.(see Figure 4 below for some samples)

Prompt: "A photo of the face of a trucker, a person"

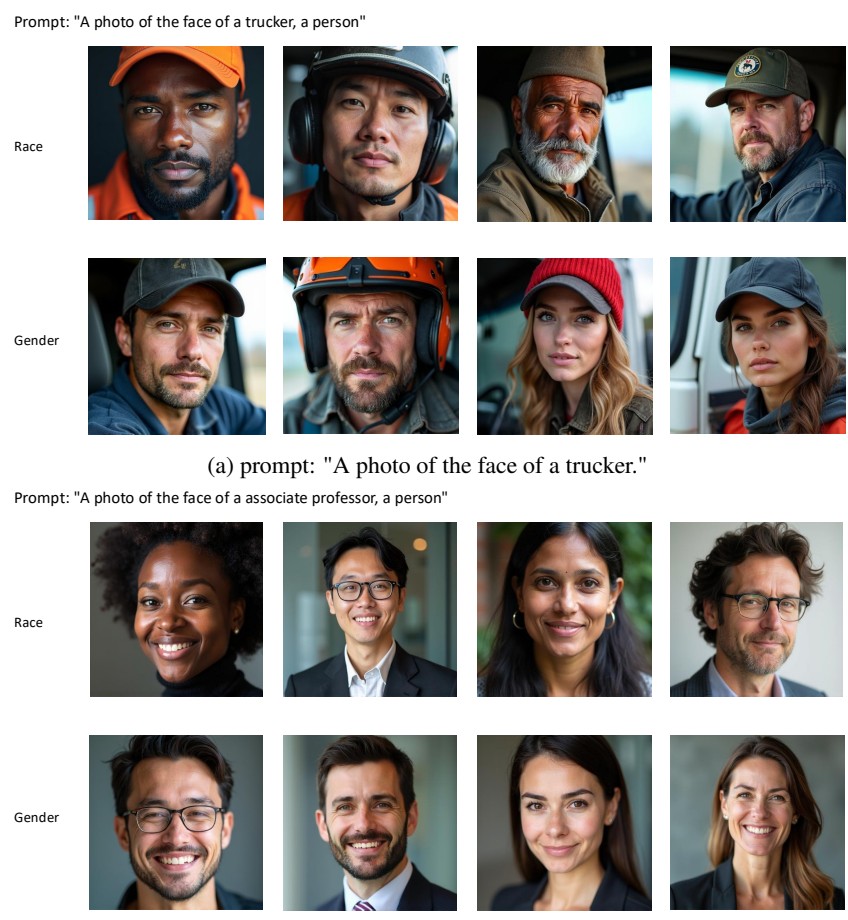

(a) prompt: "A photo of the face of a trucker."

Prompt: "A photo of the face of a associate professor, a person"

(b) prompt: "A photo of the face of an associate professor."

Figure 9: Training samples for fine-tuning. In T2I, each image is paired with its prompt (image, prompt), while in I2T, each prompt is paired with its image (prompt, image). For BPO, each prompt is paired with two images from different demographic group as (prompt, image1, image2).

# I Can debiasing understanding help debias generation?

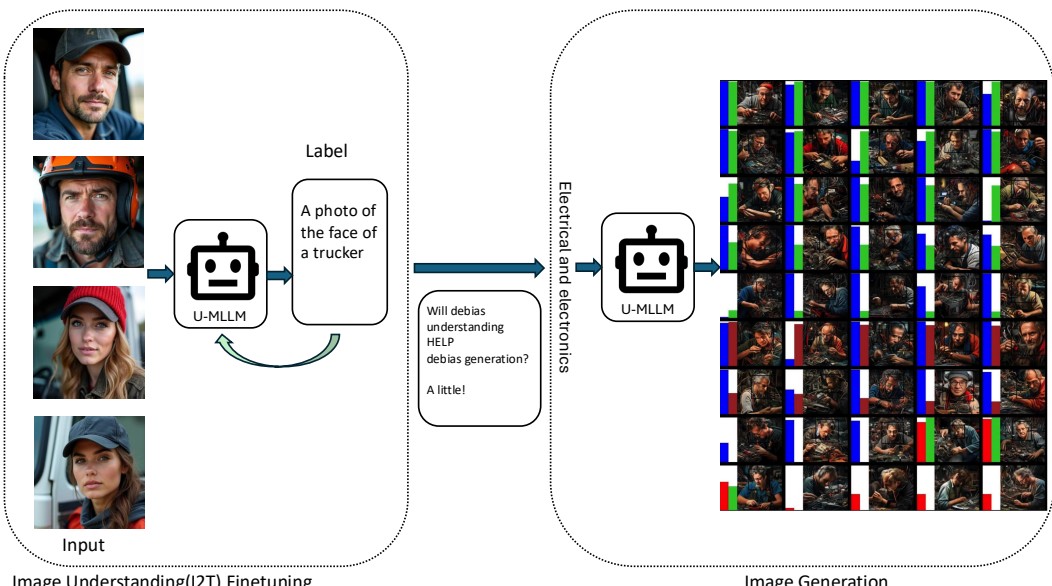

Figure 10: **Left**: We use a balanced dataset for image understanding (I2T) finetuning to debias understanding by pairing images from different demographics (e.g., female truck driver and male truck driver) with the same caption "truck driver" to teach the model that both map to the same concept. **Right**: The fine-tuned model is then used to generate an image; for each image, the first color-coded bar represents the predicted gender: male or female, and the second bar represents the predicted race: White, Asian, Black, or Indian. The height of each bar indicates the prediction confidence. **Our hypothesis is that if the model recognizes that both male and female truck drivers are truck drivers, it will generate other occupation-related concepts fairly**. However, while debiasing understanding slightly reduces bias (e.g., gender bias decreased from 0.89 to 0.83 in paper Table 1), the effect on image generation is limited. We note that recognizing multiple demographic possibilities does not alter the model's default token sampling, and without explicit training for balanced generation (e.g., T2I finetuning or Balanced Preference Optimization), biases persist.

## J Experiment Details

**Evaluation Protocol.** First, we generate $N = 160$ images for each test prompt. Next, we apply a demographic classifier from a previous study [34] to predict the labels of demographic attributes for each image. These labeled attributes are then used to compute the overall bias and semantic score based on the above metrics.

**Training Procedure.** *Base Learning Rate*: We start with a fixed learning rate (for example, $1 \times 10^{-4}$) for fine-tuning. *Batch Size*: Ranges from 8 to 32, depending on the setup. *Number of Steps*: For the first stage, we fine-tune for up to 10 epochs, checking bias and quality after the training. For the second stage, the epoch is chosen to be 1 to 2. We use the LoRA [16] method for finetuning; the rank is 32 for all experiments.

**Computational Cost** Our BPO approach starts with standard supervised fine-tuning (SFT) and adds a brief 1–2 epoch preference-optimization stage, with minimal gradient updates. Compared to other methods, supervised debiasing requires only 1–2 extra epochs to balance, while preference-learning approaches typically rely on a large reference or separate reward model; by contrast, BPO uses just a single policy. Empirically, based on training logs using two A100 GPUs (40 GB), the fine-tuning stage takes 23,020 s for gender-only, 23,020 s for race-only, and 23,193 s for mixed race–gender configurations, while the BPO stage takes roughly 2,365 s, 2,375 s, and 2,988 s, respectively. In summary, BPO is efficient and comparable in overhead to other debiasing and preference-learning methods.

# K Compare output from different methods

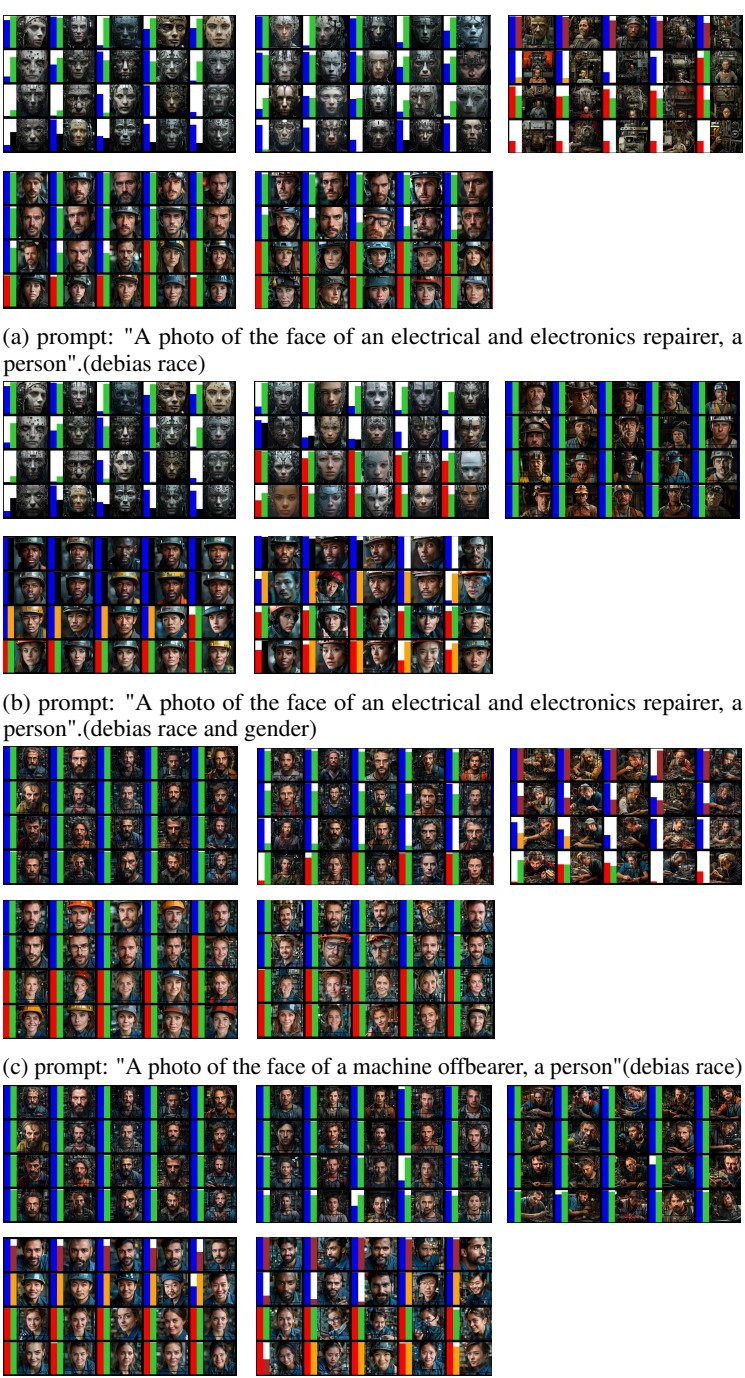

(a) prompt: "A photo of the face of an electrical and electronics repairer, a person".(debias race)

(b) prompt: "A photo of the face of an electrical and electronics repairer, a person".(debias race and gender)

(c) prompt: "A photo of the face of a machine offbearer, a person"(debias race)

(d) prompt: "A photo of the face of a machine offbearer, a person"(debias race and gender)

Figure 11: For each prompt, the first row (from left to right) shows images of VILA-U, Prompt Engineering, and I2T finetuning, while the second row shows images of T2I finetuning and BPO.

