# OpenReview forum: "On Fairness of Unified Multimodal Large Language Model for Image Generation"
_NeurIPS.cc/2025/Conference — NeurIPS 2025 poster_

### Official Review · Reviewer_iJBG · 2025-06-25

**Clarity:** 3
**Significance:** 2
**Originality:** 2
**Rating:** 4
**Confidence:** 2

**Summary:**

The authors focus their work on identifying sources of bias within a unified multimodal large language model (U-MLLMs), and implementing a “locate-then-fix” framework. The potential sources are hypothesised to have originated in the Vision Encoder and/or the Language Model. The findings presented show that the bias is likely from the language component, primarily. This is addressed with two stage debiasing approach:

1. Fine-tuning the model on a balanced dataset
2. Applying a balanced preference loss inspired by direct preference optimisation

Demographic bias mitigation is achieved while preserving semantic fidelity and image quality. This is demonstrated with improved fairness (as defined by the generation of a balanced output) while preserving the quality of the image generation.

**Questions:**

The first set of questions relate to the lack of transparency and disclosure of the use of humans in the evaluation process:
1. Why have the authors indicated that details related to the involvement of human annotators is inapplicable in the Checklist?
2. Given the involvement of annotators, and therefore, human participation, why was IRB approval not sought?

The second set of questions relate to a known phenomenon of bias stacking and amplification [1]

3. Given the known concerns of bias stacking and amplification - where models amplify biases from “upstream”, is it enough to only fix the biases in either the vision encoder or the language model?

**References:**

[1] Hall, Melissa, et al. "A systematic study of bias amplification." arXiv preprint arXiv:2201.11706 (2022).

**Ethical Concerns:**

["Major Concern: Improper research involving human subjects"]

**Final Justification:**

The authors were open to the comments in the rebuttal and showed a commendable engagement with the feedback. I believe this new score reflects the technical contributions, in line with my own assessment and the comments made by other reviewers.

**Limitations:**

1. Have the authors reflected on the limitations of the policy based on human preferences and how this might impact outcomes?

2. There are also known fairness issues when fine-tuning on balanced data that should be another point of reflection for the authors.

3. This work makes use of synthetic data, and imposes a balanced distribution as the “fair” outcome. Have the authors considered the implications of this with specific regards to unintended biases and distribution shifts [5]?

*Recommended reading:*

[1] Section 2.2.

[2] Section 6

[3] generally

4. I believe there are certain areas where additional biases can be introduced, and the authors should reflect carefully on these:

4a. Positionality of the authors
4b. Chosen categories, and the context in which these are relevant, and by extension how this might impact the interpretation of the outputs, and unintended biases

**References:**

[1] Berg, Hugo, et al. "A prompt array keeps the bias away: Debiasing vision-language models with adversarial learning." arXiv preprint arXiv:2203.11933 (2022).

[2] Smith, Brandon, et al. "Balancing the picture: Debiasing vision-language datasets with synthetic contrast sets." arXiv preprint arXiv:2305.15407 (2023).

[3] Blodgett, Su Lin, et al. "Stereotyping Norwegian salmon: An inventory of pitfalls in fairness benchmark datasets." Proceedings of the 59th Annual Meeting of the Association for Computational Linguistics and the 11th International Joint Conference on Natural Language Processing (Volume 1: Long Papers). 2021.

[4] Schrouff, Jessica, et al. "Mind the Graph When Balancing Data for Fairness or Robustness." arXiv preprint arXiv:2406.17433 (2024).

[5] Schrouff, Jessica, et al. "Diagnosing failures of fairness transfer across distribution shift in real-world medical settings." Advances in Neural Information Processing Systems 35 (2022): 19304-19318.

**Paper Formatting Concerns:**

I have no major concerns about formatting, however, while I recognise the limitations of space, key sections such as the related works and limitations should ideally not be relegated to the appendix. This is because of their importance in relation to the broader narrative of the work.

**Quality:**

2

**Strengths And Weaknesses:**

# Strengths

Well motivated paper, with a clear identification of a potential source of unfairness in  models that are under-investigated - there is a known issue of bias stacking and amplification when grouping models in a system.

# Weaknesses

I have outlined some concerns below under “Questions” and Limitations. While it is commendable that the authors are pursuing a project in fairness, I’d like to use this space to raise a concern about the vocabulary used in the paper to refer to sensitive attributes.

1. The authors are advised to carefully consider their use of gendered and racial language throughout the paper. The use of 'female' and 'male'' carry very specific assumptions and can be seen as problematic in the scope of gender recognition. One should be very careful about what is being classified and within which hegemony one makes these assumptions. Specifically, the authors are classifying attributes of a perceived gender presentation and cannot make assumptions Additionally, careful disclaimers about purely focussing on binary gender (assuming within a Western hegemony of gender, with a strong motivation for this approach are important to maintain ethical standards. See [1] for an example of this, and I strongly suggest the authors review the suggestions for further reading
2. Further, the use of discrete racial categories is also limited and the authors are encouraged to carefully consider the contexts within these categories are relevant, and the implications of these discrete categories


## Minor comments
1. This is a nitpick, but consider ordering the mini batches of references chronologically
2. Typo on L216. It should potentially read “..., we include a diffusion model”
3. Typo in the caption of Figure 4: should potentially read “*White*, Asian, Black of Indian

**References:**

[1] Hall, Siobhan Mackenzie, et al. "Visogender: A dataset for benchmarking gender bias in image-text pronoun resolution." Advances in Neural Information Processing Systems 36 (2023): 63687-63723.


**Suggested further reading:**
- [Harms of Gender Exclusivity and Challenges in Non-Binary Representation in Language Technologies](https://aclanthology.org/2021.emnlp-main.150.pdf)
- [You Keep Using That Word: Ways of Thinking about Gender in Computing Research](https://ironholds.org/resources/papers/gender_multiplicity.pdf)
- [VisoGender: A dataset for benchmarking gender bias in image-text pronoun resolution](https://arxiv.org/abs/2306.12424)
- [The Misgendering Machines: Trans/HCI Implications of Automatic Gender Recognition](https://dl.acm.org/doi/10.1145/3274357)

---

> ### Author Rebuttal · Authors · 2025-07-25
>
> Dear Reviewer iJBG,
>
> We sincerely appreciate your time and effort in reviewing our work and providing valuable feedback! Below, we address your concerns point by point.
>
>
> ---
>
> ### **Weaknesses**
>
> **1. Gendered/Racial Terminology**
>
> We deeply appreciate the reviewer's critical insight regarding our use terminology. This is indeed a fundamental concern that we take very seriously.
>
> To clarify our approach: We adopt the exact binary "male" and "female" labels from the established text-to-image diffusion fairness benchmark [1] to ensure consistence. However, we fully acknowledge that these terms represent **perceived gender presentation** in generated images rather than gender identity, and this distinction is crucial.
>
> We will add a disclaimer and modify our limitation section to acknowledge how binary classification can ignore gender-diverse individuals
>
> ---
>
> **2. Discrete Racial Categories**
>
> We acknowledge the reviewer's concern about the limitations and potential harms of discrete racial categorization. Our racial category is directly inherited from the benchmark [1]:
>
> - **WMELH** (White, Middle Eastern, Latino/Hispanic)
> - **Asian** (East Asian, Southeast Asian)
> - **Black**
> - **Indian** (South Asian)
>
> We recognize several critical issues with this taxonomy:
>
> - It reflects Western-centric constructions of race that may not be meaningful in other contexts
> - It cannot represent mixed-race individuals
>
> In our revision, we will:
>
> - Add a note explaining we use these categories solely for benchmark compatibility
> - Acknowledge that these categories are social constructs with limited validity
>
> ---
>
> ### **Questions**
>
> **1. Human Annotators Checklist Error**
>
> We sincerely apologize for this oversight in our checklist. To provide complete transparency:
>
> Our human evaluation involved only 3 adult participants who:
>
> - Were recruited as volunteers and provided informed verbal consent to participate
> - Were asked to: (a) provide occupation-based text prompts, and (b) rate generated images on a 1-5 scale for demographic diversity and image quality
> - Were not exposed to any harmful or sensitive content (only AI-generated professional portraits)
>
> We mistakenly marked this as "N/A" because we initially considered this minimal-risk evaluation of our model's outputs rather than formal human subjects research. We now recognize this was an error in judgment, and any human involvement should have been properly disclosed.
>
> ---
>
> **2. IRB Approval**
>
> We acknowledge the reviewer's valid concern about the lack of formal IRB review. Our reasoning at the time was:
>
> - The evaluation involved minimal risk (3 participants only viewed and rated AI-generated images)
> - The task was similar to routine user feedback on model outputs and no personal data was collected beyond ratings
> - Participants were colleagues who volunteered with full knowledge of the research purpose
>
> However, we now recognize that this reasoning was insufficient. We apologize for this oversight. We commit to adding explicit statements about informed consent and data handling.
>
> ---
>
> **3. Bias Amplification**
>
> This is an excellent point about cascading bias effects. Our investigation revealed that the language model was the primary source of observed bias. Our current results show:
>
> - Linear probing of vision encoder embeddings achieves high accuracy in demographic classification, confirming these attributes are encoded
> - However, when we compare token distributions between neutral and demographic-augmented prompts, the language model shows 99.80% alignment with its implicit biases
> - After applying BPO to the language model alone, overall generation bias drops significantly
>
> This suggests that while the vision encoder preserves demographic information, the language model's sampling process is where biased preferences manifest most strongly.
>
> Nevertheless, we acknowledge potential interaction effects:
>
> - The vision encoder's representation might "prime" certain token sequences
> - Subtle biases in visual features could compound with language model biases
> - Complete debiasing might require addressing both components simultaneously
>
> We will expand our discussion to include these nuances and cite relevant work suggested.
>
> ---
>
> ### **Limitations**
>
> **1. Policy Based on Human Preferences**
>
> We acknowledge that our assumption of uniform distribution as "fair" encodes specific value judgments. This approach:
>
> - Aligns with common fairness metrics in literature (demographic parity)
> - Matches the evaluation protocol of our benchmark [1]
> - Received positive feedback in our (limited) human evaluation
>
> However, we recognize this is just one possible operationalization of fairness:
>
> - It may not suit all contexts
> - Different communities might prefer proportional representation matching local demographics
> - Some applications might require context-dependent fairness criteria
>
> We will incorporate those paper suggested such as[2][3] and insights about the ethical implications.
>
> ---
>
> **2. Fine-Tuning on Balanced Data Issues**
>
>  We acknowledge that balanced training data may not ensure absolute fairness. Issues might be:
>
> - **Unnatural associations**: Equal representation might create unrealistic scenarios
> - **Downstream amplification**: Balanced training doesn't prevent bias reemergence during deployment
>
> Our approach partially addresses these through the balanced preference loss, which:
>
> - Operates on generation probabilities rather than data frequencies
> - Penalizes any systematic preference regardless of training distribution
>
> This means we're optimizing for equal likelihood of generating different demographics. The model learns to be indifferent to demographic choices at the generation level.
>
> However, we acknowledge residual risks might remain, and we will add discussion of these nuanced fairness considerations and cite relevant work suggested.
>
> ---
>
> **3. Synthetic Data and Distribution Shifts**
>
> We appreciate the reviewer's concern about potential artifacts from synthetic data. Our approach addresses these challenges through design and validation:
>
> **Why Synthetic Data is Necessary**:
>
> - **Data Scarcity**: No existing large-scale dataset provides occupational images balanced across gender × race intersections. Creating one would require photographing thousands of individuals across all demographic combinations—prohibitively expensive and prone to human annotator biases.
> - **Controlled Distribution**: Synthetic generation enables perfect demographic balance (exactly one image per demographic per prompt), which is very challenging with real-world collection.
> - **Standard Practice**: Latest U-MLLMs like Janus-Pro incorporate ~72M synthetic samples. We follow this established approach while adding targeted bias mitigation.
>
> **How We Mitigate Synthetic Artifacts**
>
> 1. **BPO Design**: Our balanced preference loss operates on token-generation probabilities. This abstraction layer insulates the debiasing objective from FLUX-specific visual patterns—we optimize for equal likelihood of generating different demographics.
> 2. **Enumeration Strategy**: By explicitly generating each demographic variant, we prevent FLUX's implicit biases from skewing the training distribution. Even if FLUX renders certain groups with characteristic styles, our balanced pairing ensures equal representation. **(see figure 8 and 9 under Appendix J)**
>
> **Empirical Evidence of Robustness(see line 247)**
>
> - **Cross-Domain**: 564 stereotype prompts beyond occupations show consistent bias reduction
> - **Cross-Language**: Chinese and French evaluations maintain effectiveness
> - **Human Validation**: User studies confirm improved diversity with minimum quality degradation
>
> While our approach proves effective, we recognize:
>
> - FLUX's aesthetic choices may subtly influence generated styles
> - Synthetic-to-real domain gap could affect deployment
> - Unknown biases in FLUX's training data might propagate
>
> We will add to our limitations section that investigating mixed synthetic-real training data could further improve robustness.
>
> ---
>
> **4. Additional Bias Sources**
>
> **4a. Author Positionality**:
> We acknowledge that our perspectives as researchers inevitably shape this work:
>
> - Our choice to focus on gender and race (rather than age, disability, etc.)
> - Our interpretation of "fairness" as demographic parity
> - Our methods for evaluation and validation from prior work[1]
>
> These choices reflect our academic training and the specific research context we operate within. Other communities might approach these questions entirely differently.
>
> **4b. Category Context and Implications**:
> The choice of gender/race as primary axes of analysis:
>
> - Following standard in bias analysis literature[1]
> - May inadvertently suggest these are the only or most important biases
> - Could ignore intersectional effects or other forms of discrimination
>
> We will add explicit acknowledgment that our categories are contextual and fairness extends  beyond the specific metrics we optimize.
>
> ---
>
> **Ethical Concerns:** We want to assure that participants were treated ethically: they volunteered for the task and were not exposed to any harmful content or risk.
>
> ---
>
> **Paper Formatting:**  We appreciate the suggestion. In revised version, We will fully integrate Related Work and Limitations into main text.
>
> ---
>
> **Reference**
>
> [1] Shen, X., et al. (2024). *Finetuning Text-to-Image Diffusion Models for Fairness*. In *Proceedings of the International Conference on Learning Representations (ICLR 2024)*.
>
> [2] Berg, Hugo, et al. "A prompt array keeps the bias away: Debiasing vision-language models with adversarial learning." arXiv preprint arXiv:2203.11933 (2022).
>
> [3] Smith, Brandon, et al. "Balancing the picture: Debiasing vision-language datasets with synthetic contrast sets." arXiv preprint arXiv:2305.15407 (2023).
>
> ---
>
> We hope these clarifications address your concerns. We are grateful for your valuable suggestions and welcome any further feedback!
>
> The author

---

> > ### Comment · Reviewer_iJBG · 2025-08-03
> > **Thank you for the detailed response**
> >
> > Dear authors
> >
> > Thank you very much for the detailed response to my review. I acknowledge the effort and reflection that went into it. Unfortunately, my concerns around the involvement of human participants remain, despite the strengths of the technical work, which the other reviewers have correctly highlighted, in my opinion. *Was an IRB approached at all (i.e. did an IRB deem this "low-risk") or was this assessment made by the team?*
> >
> > Regretfully, I do not believe my score will change, unless an IRB was consulted. While I appreciate the authors fully disclosing the involvement of the volunteers, risks posed to reviewers can also be subtle and the ethics review process beforehand should be standard - for all research teams and papers - in my opinion.

---

> ### Author Response · Authors · 2025-08-04
> **Response to Reviewer's IRB Concern:**
>
> We sincerely thank the reviewer for raising this important point about IRB approval for human evaluation. **Upon careful consideration and consultation with our institution's IRB office, we have made the decision to remove the tiny scale human evaluation component from our paper entirely**.
>
> **We acknowledge that proper IRB approval should have been sought before conducting any evaluation involving human participants, regardless of scope or risk level. We take research ethics seriously and have learned from this experience**.
>
> **Impact on Our Contributions:**
> The removal of the human evaluation section does not affect our core technical contributions or conclusions. **To put this in perspective: our human evaluation involved only 3 participants providing 3 prompts each (9 prompts total)**, while our primary evaluation uses hundreds of prompts from established benchmarks, including:
> - 1,000 occupational prompts for training
> - 50 occupational prompts for testing, 160 images per prompt
> - 564 stereotype prompts across diverse domains (personal attributes, activities, healthcare, education)
>
> Our method's effectiveness in reducing demographic bias is comprehensively demonstrated through:
>
> 1. **Quantitative Metrics**:
>    - 72% reduction in gender bias (0.89 → 0.25)
>    - 46% reduction in race bias (0.48 → 0.26)
>    - Consistent improvements across all prompt categories
>
> 2. **Technical Validation**: Our component-wise analysis and ablation studies provide strong evidence for the effectiveness of our Balanced Preference Optimization (BPO) approach
>
> 3. **Generalization Results**: Cross-language evaluation (Chinese, French) and out-of-distribution prompts demonstrate robust bias reduction
>
> The human evaluation was icing on the cake—a tiny supplementary component to our total evaluation. All conclusions in the paper are supported by our extensive computational experiments and automated metrics, which is from existing study in the fairness evaluation benchmarks[1].
>
> **Nevertheless, we sincerely apologize for this oversight**. To provide complete transparency: We will revise the manuscript to remove all references to human evaluation while maintaining the comprehensive technical evaluation that forms the foundation of our contributions. **Additionally, we are working with our IRB office to ensure full compliance with research ethics protocols for any future work involving human participants.**
>
> We hope this addresses the reviewer's ethical concerns while preserving the scientific rigor and contribution of our work. We are sincerely grateful for your guidance in helping us meet the highest ethical standards in our research!
>
> ---
>
> Reference
>
> [1] Shen et al. (2024). Finetuning Text-to-Image Diffusion Models for Fairness. ICLR 2024.
>
> ---
>
>
> The author

---

> > ### Comment · Reviewer_iJBG · 2025-08-05
> > **The authors are commended for their engagement with the review and efforts to consult their IRB**
> >
> > Dear Authors
> >
> > I have a lot of respect for how this has been handled this and I commend the time taken to carefully reflect on the points raised. I believe the lesson has been learnt, and I also genuinely hope this practice of consulting IRBs for any scale of human annotation work becomes more ingrained and non-negotiable in the broader field.
> >
> > In response to the author's efforts, I am increasing my score to 4 to reflect the technical contribution, however I do hope the authors take serious steps to updating the "Gendered/Racial Terminology". As a field, we need to be more sensitive to avoid creating "frozen moments in time" [1,2] and severely inhibit progress for representation of more marginalised groups.
> >
> > All the best with the rest of the process!
> >
> > **References**
> >
> > [1] Shreya Shankar, Yoni Halpern, Eric Breck, James Atwood, Jimbo Wilson, and D Sculley. 2017. No classification without representation: Assessing geodiversity issues in open data sets for the developing world. arXiv preprint arXiv:1711.08536
> > (2017).
> >
> > [2] Laura Weidinger, John Mellor, Maribeth Rauh, Conor Griffin, Jonathan Uesato, Po-Sen Huang, Myra Cheng, Mia Glaese, Borja Balle, Atoosa Kasirzadeh, et al. 2021. Ethical and social risks of harm from language models. arXiv preprint arXiv:2112.04359 (2021).

---

> ### Author Response · Authors · 2025-08-05
>
> Dear Reviewer,
>
> We are very grateful for your positive reassessment and support! Your feedback has been invaluable to us. We completely agree with your concerns regarding the "Gendered/Racial Terminology" and are committed to making the necessary revisions in our final version.
>
> Thank you for helping us improve our work!
>
> The author

---

### Official Review · Reviewer_3uNx · 2025-06-30

**Clarity:** 4
**Significance:** 3
**Originality:** 3
**Rating:** 5
**Confidence:** 5

**Summary:**

This paper presents an analysis and debiasing technique targeting social biases in unified multimodal large language models (U-MLLMs). Biases are analysed by probing the image tokens the model autoregressively generates and by comparing their distributions. For debiasing, the paper proposes a preference tuning method based on using synthetic images. Furthermore, the paper analyses biases in understanding and generation and their relation, and investigates additional loss formulations for debiasing.

**Questions:**

- Why does the evaluation only consider occupational prompts and not further topics as well?
- Can we apply the debiasing method in scenarios where we aim for a skewed target distribution of attributes?
- How were the race labels selected and why?

**Ethical Concerns:**

["NO or VERY MINOR ethics concerns only"]

**Final Justification:**

The presented analysis of biases and debiasing techniques in U-MLLMs is both novel and interesting. It is especially interesting that the paper develops methods specifically targeted at U-MLLMs. All remaining concerns were addressed in the rebuttal, and all reviewers expressed a positive evaluation of this paper.

**Limitations:**

See Weaknesses.

**Paper Formatting Concerns:**

No paper formatting concerns

**Quality:**

3

**Strengths And Weaknesses:**

## Strengths
**(S1)** A targeted analysis of biases and debiasing techniques in U-MLLMs is a novel and very interesting perspective to study. This paper also clearly develops unique approaches that are different compared to existing approaches to studying social bias in Diffusion Models, for example the image token distribution and bias in generation vs. understanding.

**(S2)** Using the image token distribution to assign demographic labels (instead of external models that predict this information from the generated images) seems like a great idea. This potentially allows for analyzing bias without even assigning explicit labels, which can be controversial [a]. Instead, bias analysis would rely on the distributions, similar to [b]. Further analyses of what the image tokens encode (e.g. are we really measuring gender representations in the tokens and not correlated attributes such as clothing spurious correlations in background patterns) and whether this technique could also be applied to other models that use VQ-VAEs (such as Diffusion Models) would be greatly appreciated.

**(S3)** The paper discusses the relation of bias in understanding (text output) and generation (image output) scenarios. This is a unique and important area of investigation for U-MLLMs. Future work has to explore this relation and possible strategies to transfer bias mitigation between modalities further, but the conducted experiments are a good start.

**(S4)** The paper contains a human evaluation of debiasing success and ablations for a number of loss formulations. These additional experiments are appreciated.

## Weaknesses
**(W1)** The evaluation focuses on occupational prompts. While this is standard in bias analysis and debiasing literature, a greater variety of prompts would strengthen the validity of the present analysis. The experimental setup doesn't have to be limited to occupations. The main advantage of using occupation is that it allows easy exploration of bias amplification by comparing the resulting proportions (e.g. of male-/female-gendered images) to real-world statistics, but the present paper does not make use of this. Therefore, other prompt topics can be integrated without causing any problems.

**(W2)** The paper only considers parity (uniform distribution over attributes) as debiasing target. However, this is a strong limitation. In practice, it should be possible to adapt models to arbitrary target distributions, as there is no consensus on whether uniform distribution should be the standard. For example, the "Gemini incident" (where a proprietary model produces images of Vikings with dark skin or images of nazi soldiers with Asian features) demonstrated that uniform distributions do not make sense in many contexts. The present paper would benefit from adding a discussion on how the proposed methods can be used to adapt to arbitrary sampling distributions and how to maintain contextual appropriateness.

**(W3)** The limitations section does not include a discussion of the chosen gender/race categories. However, it would be greatly appreciated if the paper could include a discussion on potential problems with discrete labels for categories that inherently lie on a spectrum. Especially, race is a problematic case, as there is little consensus on which are appropriate labels (see [a|).

## Typos and Minor Weaknesses
* Line 60: It is unclear here what the bias values mean, as they haven't been introduced yet
* Line 463: The related work section mentions "few studies" that explored bias in U-MLLMs, but cites none. This asks whether after all there are any previous studies on this or not.

## Justification for Rating
The evaluation is limited, focusing on occupational prompts only. Also, the debiasing methods targets uniform distribution over attributes and not arbitrary target distributions. Nonetheless, the value of providing solid insights and unique methods to the study of social bias in U-MLLMs, which is a clearly novel area of research, outweighs my concerns regarding the evaluation. I would like to encourage development of novel evaluation and debiasing approaches for this relevant type of image generation, and this paper provides a strong starting point.

## References
[a] Khan & Fu (2021): One Label, One Billion Faces: Usage and Consistency of Racial Categories in Computer Vision. In FAccT\
[b] Luccioni et al. (2023): Stable Bias: Evaluating Societal Representations in Diffusion Models. In NeurIPS

---

> ### Author Rebuttal · Authors · 2025-07-26
>
> Dear Reviewer 3uNx,
>
> We sincerely appreciate your time and effort in reviewing our work and providing valuable feedback! Below, we address your concerns point by point.
>
>
> ---
>
> ### **(W1) Limited evaluation to occupational prompts**
>
> We appreciate the reviewer's observation and want to clarify that our evaluation extends  beyond occupational prompts. As mentioned briefly in line 247, we conduct extensive testing on **564 stereotype prompts** from the latest bias benchmark[3], covering diverse domains:
>
> - **Personal Attributes**: "A face of a kind person," "A face of an honest individual"
> - **Daily Activities**: "A face of a person folding freshly washed clothes"
> - **Healthcare Contexts**: "A face of an individual with diabetes"
> - **Educational Settings**: "A face of a sophomore"
>
> These non-occupational prompts constitute a substantial portion of our evaluation and demonstrate consistent bias reduction across contexts. We will add more details to better highlight these results and show that our method generalizes beyond occupations. We agree that occupational bias analysis often compares to real-world statistics—while we don't perform this comparison in this work, our broader evaluation validates the method's general applicability.
>
> ---
>
> ### **(W2) Limitation to uniform distribution target**
>
> This is an excellent point that touches on fundamental questions about fairness definitions! We acknowledge that uniform distribution is indeed one specific solution of fairness, not a universal solution.
>
> **Current Approach**: Our BPO loss minimizes pairwise odds ratios to achieve demographic parity, which aligns with widely-used fairness metrics in the literature and our evaluation benchmark[1].
>
> **Extension to imbalanced Distributions**: Our framework might be adapted to non-uniform targets by modifying the loss function. Instead of having |OR(y_di, y_dj) - 1| in our loss function, we could have |OR(y_di, y_dj) - τ_ij| where τ_ij represents the desired ratio between demographics di and dj.  We will add discussion addressing:
>
> 1. The Gemini incident as a cautionary example of context-blind uniformity
> 2. The trade-off between fairness and contextual accuracy
>
> ---
>
> ### **(W3) Missing discussion on categorical limitations**
>
> We fully agree that our use of discrete demographic categories is a significant limitation that warrants thorough discussion. We will add more details in Limitations:
>
> **Gender Categories**: We adopt binary labels ("male"/"female") from the established benchmark [1] to ensure comparability. However, we explicitly acknowledge:
>
> - These represent **perceived presentation** in images, not gender identity
> - This binary framework excludes non-binary and gender-diverse individuals
>
> **Race Categories**: Our four-group taxonomy (WMELH, Asian, Black, Indian) directly follows [1]:
>
> - **WMELH**: {White, Middle Eastern, Latino/Hispanic}
> - **Asian**: {East Asian, Southeast Asian}
> - **Black**: African and African diaspora
> - **Indian**: South Asian
>
> We acknowledge this categorization:
>
> - Conflates distinct ethnicities (problematic as noted in [2])
> - Reflects Western-centric racial constructs
> - Cannot capture mixed-race individuals
>
> We will emphasize that we use these categories solely for benchmark compatibility while recognizing their fundamental limitations in representing human diversity.
>
> ---
>
> ### **Minor Issues**
>
> Thanks for the suggestions!
>
> **Line 60**: We will add a parenthetical explanation: "where 0 indicates perfect balance and 1 indicates complete skew toward one gender".
>
> **Line 463**: We will correct this by Revising the sentence.
>
> ---
>
> ### **Response to Questions**
>
> **Q1: Why only occupational prompts?**
>
> As clarified above under w1, we actually test on 564 diverse prompts beyond occupations. We will restructure our presentation to make this clearer. Occupations serve as our primary training domain because they're well-studied in bias literature, but our evaluation demonstrates broader applicability.
>
> **Q2: Skewed target distributions?**
>
> Yes, our method might be adapted. The balanced preference loss could be modified to target imbalanced distribution by replacing the uniform odds ratio target (OR=1) with custom ratios. Specifically:
>
> ```
> L_custom = Σ log[1 + σ(log OR(y_di, y_dj) - log τ_ij) - 0.5]²
>
> ```
>
> where τ_ij encodes the desired probability ratio p(di)/p(dj). We will add this in discussion.
>
> **Q3: Race label selection?**
>
> We inherit the exact categorization from the established fairness benchmark [1] to ensure consistence. We recognize the limitations of these categories (as discussed in W3). We will add a thorough discussion of these limitations and encourage future work to develop more nuanced demographic representations.
>
> ---
>
> **Reference**
>
> [1] Shen, X., et al. (2024). *Finetuning Text-to-Image Diffusion Models for Fairness*. In *Proceedings of the International Conference on Learning Representations (ICLR 2024)*.
>
> [2] Khan & Fu (2021): One Label, One Billion Faces: Usage and Consistency of Racial Categories in Computer Vision.
>
> [3] Xu, C., et al. (2025). MMDT: Decoding the Trustworthiness and Safety of Multimodal Foundation Models. In *Proceedings of the International Conference on Learning Representations (ICLR 2025)*.
>
> ---
>
> We sincerely thank the reviewer for these thoughtful critiques, which will substantially strengthen our paper's contribution and clarity.
>
> The author

---

> > ### Comment · Reviewer_3uNx · 2025-08-01
> >
> > Thank you for the detailed answer. I have read all reviews and the respective rebuttals. I will take them into consideration when finalizing my rating.
> >
> > Overall, I think the rebuttals are on point and provide important clarifications.
> >
> > > (W1) Limited evaluation to occupational prompts
> >
> > I acknowledge this was a misunderstanding on my part. However, I would like to point out that all examples and mentions of prompts in the paper only concern occupation prompts. This is also the case for Shen et al. (2024), which inspired the training prompts. I strongly encourage the authors to be more explicit about non-occupation prompts, e.g., by giving examples around line 247 and including qualitative examples in the supplementary material. This would clarify this aspect to readers who are not familiar with reference [44].
> >
> > Shen et al.: Finetuning Text-to-Image Diffusion Models for Fairness. ICLR 2024
> >
> > ---
> >
> > Currently, I'm leaning towards raising my rating to 5. I think this is an interesting work, and all concerns have been addressed in the rebuttals. I strongly encourage including in the revision the clarifications about demographic categories and the human study as provided in the rebuttals. These are relevant and important aspects.

---

> > > ### Author Response · Authors · 2025-08-01
> > >
> > > Dear Reviewer 3uNx,
> > >
> > > Thank you for your helpful suggestions! We promise to implement your recommendations in the final version. We will clarify the prompt scope and integrate the clarifications from the rebuttal. Really appreciate your support!
> > >
> > > The author

---

### Official Review · Reviewer_GpU1 · 2025-07-02

**Clarity:** 2
**Significance:** 2
**Originality:** 2
**Rating:** 4
**Confidence:** 3

**Summary:**

This paper investigates demographic bias (specifically gender and race) in Unified Multimodal Large Language Models (U-MLLMs), which are capable of both visual understanding and generation. The authors first benchmark several state-of-the-art U-MLLMs, demonstrating that they exhibit significant demographic biases in their image generation outputs, often amplifying stereotypes present in the training data. To pinpoint the origin of this bias, the paper introduces a "locate-then-fix" framework. Through a series of audits, including linear probing of vision encoder embeddings and analyzing the token distribution of the language model, the authors hypothesize that the language model component is the primary source of generative bias. The authors also identify a "partial alignment" phenomenon, where models may appear less biased in text-based understanding tasks (e.g., VQA) but still produce demographically skewed images. To mitigate these issues, the paper proposes a two-stage debiasing method. First, it synthesizes a demographically balanced dataset using a text-to-image diffusion model. Second, it introduces a novel balanced preference optimization (BPO) loss function, inspired by Direct Preference Optimization (DPO), which encourages the model to assign equal generation probabilities across different demographic groups for neutral prompts. Extensive experiments on models like VILA-U show that the proposed BPO method significantly reduces gender, race, and intersectional bias while preserving, and in some cases improving, image quality and semantic fidelity.

**Questions:**

1. Regarding the "Partial Alignment" Phenomenon: You compellingly show that debiasing understanding (I2T finetuning) has only a marginal effect on debiasing generation. You connect this to the "Reversal Curse". Could this gap also be explained by the idea that the "understanding" and "generation" tasks operate on different levels of abstraction within the model? That is, understanding maps diverse inputs to a single concept, while generation maps a single concept to a specific, high-probability token sequence that reflects the training data's mode.

2. On the Choice of Data Synthesizer: You use FLUX to generate the balanced dataset. Did you consider or experiment with other generators? Could the choice of generator significantly impact the effectiveness of the BPO fine-tuning, for instance, if a different generator had a stronger or weaker intrinsic bias for certain features correlated with demographics?

**Ethical Concerns:**

["NO or VERY MINOR ethics concerns only"]

**Limitations:**

Please see the weakness.

**Paper Formatting Concerns:**

No concern

**Quality:**

3

**Strengths And Weaknesses:**

Strength:
1. Timely and Significant Problem: The paper addresses the critical and timely issue of fairness in the latest generation of U-MLLMs. As these models become more prevalent, understanding and mitigating their potential to amplify societal biases is of paramount importance.
2. Systematic "Locate-then-Fix" Approach: The diagnostic framework for localizing the source of bias is a major strength. Rather than treating the U-MLLM as a black box, the authors conduct separate audits of the vision and language components. The experiments using linear probing on image embeddings and Jensen-Shannon divergence on token distributions provide compelling evidence that the language model is the dominant source of generative bias.
3. Comprehensive Experimental Evaluation: The paper's evaluation is thorough and robust.

Weakness:
1. Reliance on a Synthetic Dataset from a Biased Model: The proposed solution relies on generating a balanced dataset using the FLUX model. While the authors use an enumeration strategy to enforce demographic balance, the visual characteristics and potential subtle biases of the FLUX model itself are inherited by the training data. The paper acknowledges FLUX is not bias-free, but the downstream impact of using a potentially biased tool for debiasing could be explored further.
2. Limited Scope of Demographic Attributes: The study focuses exclusively on gender (binary) and race. While this is a critical starting point, the paper acknowledges that other attributes like age, culture, religion, and disability are not addressed. The proposed BPO method requires discrete, enumerable demographic groups, and it is unclear how it would scale to more complex or continuous attributes.

---

> ### Author Rebuttal · Authors · 2025-07-27
>
> Dear Reviewer GpU1,
>
> We sincerely appreciate your time and effort in reviewing our work and providing valuable feedback! Below, we address your concerns point by point.
>
>
> ---
>
> ### **1. Reliance on Synthetic Dataset from a Biased Model**
>
> We appreciate this insightful critique. Our approach specifically addresses these limitations:
>
> **Bias Mitigation at Multiple Levels**:
>
> 1. **Enumeration Strategy**: By explicitly generating images for each demographic combination, we prevent FLUX's implicit biases from affecting the demographic distribution in our dataset. Even if FLUX generates stereotypical features for certain demographics, our balanced pairing ensures equal representation(See pipeline in Figure 8 under appendix J and samples in Figure 9).
> 2. **Loss Function Design**: Our BPO loss operates on token-generation probabilities. This means we're optimizing for equal likelihood of generating different demographics regardless of FLUX's visual biases. The model learns to be indifferent to demographic choices at the generation level.
> 3. **Empirical Evidence of Robustness(see line 247)**
>
>     **Cross-Domain**: 564 stereotype prompts beyond occupations show consistent bias reduction
>
>     As mentioned briefly in line 247, we conduct extensive testing on **564 stereotype prompts** from the latest bias benchmark[1], covering diverse domains:
>
>     - **Personal Attributes**: "A face of a kind person," "A face of an honest individual"
>     - **Daily Activities**: "A face of a person folding freshly washed clothes"
>     - **Healthcare Contexts**: "A face of an individual with diabetes"
>     - **Educational Settings**: "A face of a sophomore"
>
>     **Cross-Language**: Chinese and French evaluations maintain effectiveness.
>
>     **Human Validation**: User studies confirm improved diversity with minimum quality degradation
>
>
> ---
>
> ### **2. Limited Scope of Demographic Attributes**
>
> We appreciate the reviewer's concern and acknowledged this in our limitation section in Appendix. Our method currently requires discrete categories. We adopt binary labels ("male"/"female") and category of race from the established benchmark [2] to ensure comparability. This indeed constrains its applicability:
>
> **Current Limitations**:
>
> - Binary gender categories exclude non-binary identities
> - Discrete racial categories cannot capture mixed-race individuals or continuous ethnic variations
> - Attributes like age, disability, and culture are not addressed
>
> **Challenges for Extension**:
>
> 1. **Continuous Attributes**: For age, we could discretize into ranges (20-30, 30-40, etc.) but this loses granularity
> 2. **High-Cardinality Attributes**: For culture/nationality (200+ categories), the pairwise loss becomes computationally expensive: O(K²) comparisons.
>
> **Possible Solutions such as** Group similar demographics hierarchically. We promise to add more discussing these details.
>
> ---
>
> ### **3. "Partial Alignment" Phenomenon and Levels of Abstraction**
>
> This is an excellent insight that enriches our understanding of the phenomenon.
>
> - **Understanding (I→T)**: Maps diverse visual inputs → single abstract concept
>     - "Male professor" image → "professor" (many-to-one mapping)
>     - Operates at high semantic level, abstracting away visual details
> - **Generation (T→I)**: Maps abstract concept → specific visual instance
>     - "Professor" → specific image tokens (one-to-many mapping, different images)
>     - Sample from learned distribution
>
> **Connection to Our Findings**:
> This abstraction-level difference explains why I→T finetuning fails to debias generation:
>
> 1. During I→T training, the model learns to ignore demographic features (abstraction)
> 2. During T→I generation, the model must still select specific features, defaulting to the highest-probability (biased) mode
> 3. The "Reversal Curse" is thus a special case of this broader abstraction asymmetry
>
> **Implications**:
>
> - Debiasing needs to target the specific level where bias manifests
> - Understanding tasks may require different debiasing strategies than generation tasks
>
> Really appreciate the insights!
>
> ---
>
> ### **4. Choice of Data Synthesizer**
>
> We appreciate this insightful suggestion. Before we generated dataset, we did experiment and evaluate multiple image generators for creating our balanced dataset:
>
> **Generator Comparison**:
>
> 1. **FLUX.1-dev** (Selected):
>     - State-of-the-art at the time of our study
>     - Superior demographic control - reliably generates requested gender/race attributes
>     - Consistent high-quality outputs across most occupational categories(see samples in Figure 9)
>     - Stable generation with minimum failure cases
> 2. **Stable Diffusion v1.5** (Evaluated but not selected):
>     - Less reliable demographic adherence - often ignored specific gender/race prompts
>     - Variable quality, particularly for certain occupations
>     - Failed to generate coherent images for some professional categories (e.g., produced low quality and wrong outputs for some occupations)
>
> Our approach would work with sufficiently capable image generator that can reliably produce demographic variations. We will add these details. Thank you for the suggestion!
>
> ---
>
> **Reference**
>
> [1] Xu, C., et al. (2025). MMDT: Decoding the Trustworthiness and Safety of Multimodal Foundation Models. In *Proceedings of the International Conference on Learning Representations (ICLR 2025)*.
>
> [2] Shen, X., et al. (2024). *Finetuning Text-to-Image Diffusion Models for Fairness*. In *Proceedings of the International Conference on Learning Representations (ICLR 2024)*.
>
> ---
>
> We sincerely thank the reviewer for these thoughtful critiques, which will substantially strengthen our paper's contribution and clarity.
>
> The author

---

> > ### Comment · Reviewer_GpU1 · 2025-08-05
> > **Thank you fot the response**
> >
> > I thank the author for the detailed responses. My concerns have been sovled well and I will hold my postitive rating score and evaluation of this work. About my concerns proposed in the review, I hope authors to provide more quantitative indicators and experimental explanation in the next version.

---

> > > ### Author Response · Authors · 2025-08-05
> > >
> > > Dear Reviewer,
> > >
> > > Thank you for your positive feedback and for your support of our work. We will be sure to incorporate the additional details you suggested into the final version of the manuscript!
> > >
> > > The author

---

### Official Review · Reviewer_BVqp · 2025-07-05

**Clarity:** 4
**Significance:** 3
**Originality:** 3
**Rating:** 4
**Confidence:** 3

**Summary:**

The paper audits recent unified MLLMs and shows they generate notably gender- and race-skewed images even when their text-only answers appear unbiased. By dissecting vision tokenizer vs. LM components, the authors trace most bias to the LM and propose a two-stage “Balanced Preference Optimization” (BPO) that fine-tunes on a synthetically balanced dataset and applies an odds-ratio loss, cutting VILA-U’s gender bias by ~72 % while keeping image quality basically unchanged.

**Questions:**

1. I would like to suggest the authors to discuss more about the generalization of their training data generation pipeline.

2. What is the core difference between the proposed BPO and standard DPO?

**Ethical Concerns:**

["NO or VERY MINOR ethics concerns only"]

**Limitations:**

Yes.

**Paper Formatting Concerns:**

N/A.

**Quality:**

3

**Strengths And Weaknesses:**

- Strength

1. The paper is clear and easy-to-follow.
2. Decoupling and analyze the vision encoder and LM is a clear and feasible way to identify the bias.
3. The evaluation shows consistent improvements on a broad range of benchmarks.

- Weaknesses

1. One major weakness is that the authors use template-based training prompts (~1000), which is basically designed by hand and is hard to scale to more diverse prompts and broader types of biases.
2. Apart from the training prompts, the paper construct their training images purely with FLUX.1 instead of using real-world web images. There are risks of leaking FLUX.1's artefacts or reducing the diversity of generated images of the unified MLLMs.

---

> ### Author Rebuttal · Authors · 2025-07-26
>
> Dear Reviewer BVqp,
>
> We sincerely appreciate your time and effort in reviewing our work and providing valuable feedback! Below, we address your concerns point by point.
>
> ---
>
> ### **1. Template-based training prompts limitation**
>
> We appreciate the reviewer's concern about scalability. In fact, our approach is more diverse than it may initially appear. We promised to add more details in our pipeline figure 8 under Appendix J. We employ a multi-stage augmentation strategy:
>
> 1. **Prompt Generation**: Starting from 1,000 base templates, we use ChatGPT to generate semantically equivalent paraphrases, expanding our prompt set to over much more unique variations. For example, "an associate professor" is paraphrased into "university faculty member," "academic researcher," "college instructor," etc.
> 2. **Semantic Diversity**: This paraphrasing ensures our model encounters diverse linguistic expressions mapping to the same occupational concepts, preventing overfitting to specific phrasings and better simulating real-world prompt diversity.
> 3. **Scalability**: Our pipeline is scalable - new occupations or domains can be added and automatically expanded through the same paraphrasing process. We demonstrate generalization through our evaluation on 564 stereotype prompts (line 247) that were not seen during training. These stereotype prompts covering diverse domains:
> - **Personal Attributes**: "A face of a kind person," "A face of an honest individual"
> - **Daily Activities**: "A face of a person folding freshly washed clothes"
> - **Healthcare Contexts**: "A face of an individual with diabetes"
> - **Educational Settings**: "A face of a sophomore"
>
> ---
>
> ### **2. Synthetic FLUX data vs. real-world images**
>
> We acknowledge the reviewer's valid concern about synthetic data artifacts. Our approach specifically addresses these limitations:
>
> 1. **Necessity of Synthetic Data**: Based on our knowledge, there is no existing large scale dataset provides real-world occupational images balanced across both gender and race dimensions. Creating such a dataset manually would be prohibitively expensive and potentially introduce human annotator biases.
> 2. **Widespread Practice**: Synthetic data is standard in training unified MLLMs - For example, Janus-Pro uses ~72M synthetic samples for image generation. Our approach follows established practices while adding specific bias mitigation.
> 3. **Empirical Validation**: Our extensive evaluations demonstrate:
>     - Consistent bias reduction across different test sets (stereotype prompts, cross-language)
>     - Human evaluation confirming improved diversity with minimum degradation in quality
>     - Generalization to out-of-distribution prompts not seen during training
> 4. **Future Work**: We will add to our limitations section that investigating mixed synthetic-real training data could further improve robustness.
>
> ---
>
> ### **3. Generalization of training data generation pipeline**
>
> Thanks for the suggestion! In figure 8 under appendix J, we will expand our discussion of the data generation pipeline's generalization capabilities:
>
> 1. **Pipeline Robustness**: Our approach explicitly controls for demographic attributes through augmented prompts ("Indian associate professor," "female construction worker"), ensuring coverage across all demographic intersections. This enumeration strategy is model-agnostic and can work with any sufficiently capable image generator.
> 2. **Balanced Pairing Strategy**: Each neutral prompt is paired with one image per demographic group, creating a balanced training distribution(see Figure 9 for some samples). This guarantee of balance is independent of the underlying generator's biases.
> 3. **Generalization Evidence**:
>     - Cross-language evaluation (Chinese, French) shows consistent bias reduction
>     - Performance on unseen stereotype prompts demonstrates learned fairness principles as mentioned before
>
> ---
>
> ### **4. Core difference between BPO and standard DPO**
>
> The fundamental distinction between our Balanced Preference Optimization (BPO) and standard Direct Preference Optimization (DPO) lies in objectives and mathematical formulations:
>
> **DPO**:
>
> - **Objective**: Maximize likelihood of human-preferred responses over rejected ones
> - **Formulation**: Binary preference p(y_preferred > y_rejected | x)
> - **Use case**: General alignment with human preferences
>
> **BPO**:
>
> - **Objective**: Enforce uniform generation probability across demographic groups
> - **Formulation**: Minimize pairwise odds ratio deviations: min Σ[log(1 + σ(log OR(y_di, y_dj)) - 0.5)²]
> - **Use case**: Designed for demographic fairness
>
> **Practical Difference**: While DPO optimizes a single preference direction (good vs bad), BPO simultaneously balances pairwise preferences to achieve demographic parity. DPO would require labeling images as "preferred" or "rejected," which is subjective for preference. BPO directly optimizes for equal representation: the optimized model has no preference between generating demographic di or dj, achieving fairness without subjective preference labels.
>
> This balancing addresses the unique challenges of fairness in generation, where the goal isn't to prefer one output over another, but to ensure no systematic preference exists across demographic groups.
>
> ---
>
> We sincerely thank the reviewer for these thoughtful feedback, which will substantially strengthen our paper's contribution and clarity!
>
> The author

---

> > ### Comment · Reviewer_BVqp · 2025-08-09
> >
> > Thank the authors for their effort and response. Please revise the manuscript to include the additional details. I would keep my positive score.

---

### Decision · Program_Chairs · 2025-09-17

**Decision:**

Accept (poster)

**Comment:**

This paper investigates demographic bias (gender and race) in Unified Multimodal Large Language Models (U-MLLMs) and benchmarks several state-of-the-art U-MLLMs. Overall, the paper is clear and easy-to-follow. The paper addresses the critical and timely issue of fairness in the latest generation of U-MLLMs. All the reviewers lean to accept the paper. Therefore, I recommend accept. I also recommend the authors to follow the reviewers' comments, e.g, update the "Gendered/Racial Terminology".